

# Centennial-scale precipitation anomalies in the southern Altiplano (18° S) suggest an extra-tropical driver for the South American Summer Monsoon during the late Holocene

Ignacio A. Jara[1], Antonio Maldonado[1,2,3], Leticia González, Armand Hernández[4], Alberto. Sáez[5], Santiago Giralt[5], Roberto Bao[6], Blas Valero-Garcés [7,8]

[1]Centro de Estudios Avanzados en Zonas Áridas (CEAZA), Colina del Pino, La Serena, Chile

[2]Instituto de Investigación Multidisciplinario en Ciencia y Tecnología, Universidad de La Serena, La Serena, Chile

[3]Departamento de Biología Marina, Universidad Católica del Norte, Larrondo 1281, Coquimbo, Chile

[4]Instituto Ciencias de la Tierra Jaume Almera- CSIC, Barcelona, Spain

[5]Departament de Dinàmica de la Terra i de l'Oceà. Universitat de Barcelona, Spain

[6]Centro de Investigacións Científicas Avanzadas (CICA), Facultade de Ciencias, Universidade da Coruña, A Coruña, Spain

[7]Instituto Pirenaico de Ecología – CSIC, Zaragoza, Spain

[8]Laboratorio Internacional de Cambio Global, CSIC-PUC-UFRJ, Zaragoza, Spain

Correspondence to: antonio.maldonado@ceaza.cl


**Abstract**

Modern precipitation anomalies in the Altiplano region of South America are closely linked to the strength of the South

American Summer Monsoon (SASM) which is influenced by large-scales climate features sourced in the tropics such as

latitudinal shifts of the Intertropical Convergence Zone (ITCZ) and El Niño-Southern Oscillation (ENSO). However,

the timing, direction and spatial extent of precipitation changes prior to the instrumental period are still largely

unknown, preventing a better understanding of the long-term drivers of the SASM and their effects over the Altiplano.

Here we present a detailed pollen reconstruction from a sedimentary sequence covering the period between 4500-1000

cal yr BP in Lago Chungará (18° S; 4570 masl), a high elevation lake in the southwestern margin of the Altiplano

where precipitation is delivered almost exclusively during the mature phase of the SASM in the austral summer. We

distinguish three well-defined centennial-scale anomalies, with dry conditions between 4100-3300 and 1600-1000 cal yr

BP, and a conspicuous humid interval between 2400-1600 cal yr BP; which resulted from weakening and strengthening

of the SASM respectively. Comparisons with other climate reconstructions from the Altiplano, the Atacama Desert, the

Tropical Andes and the southwestern Atlantic coast reveal that - unlike the modern climatological controls - past

precipitation anomalies at Lago Chungará were largely decoupled from north-south shifts in the ITCZ and ENSO. A

regionally coherent pattern of centennial-scale SASM variations and a significant latitudinal gradient in precipitation

responses suggest the contribution of an extra-tropical moisture source for the SASM, with significant effects over

precipitation variability in the Southern Altiplano.

**1 Introduction**

Detailed paleoclimate records are not only necessary to document past climate anomalies, but also to decipher regional

drivers of atmospheric variability and to explore teleconnections that extend beyond the period instrumental climate

measurements (Wanner et al., 2008; Loisel et al., 2017). Unfortunately, the number of reliable climate reconstructions

is still low in many regions of the world which makes difficult to put modern climate trends into a longer-term context

of climate variability. This long-term perspective is necessary considering the brief period of meteorological

measurements, the current scenario of unprecedented climate change, and the large uncertainties about future trends

(Deser et al., 2012; Neukom et al., 2015).

In the tropical Andes (5-13° S) and the Altiplano (13-22° S) regions of South America, a set of well-grounded

reconstructions have revealed that significant precipitation changes occurred during the most recent millennia (Seltzer

et al., 2000; Baker et al., 2005; Giralt et al., 2008; Bird et al., 2011a; Novello et al., 2016), reflecting decadal,

centennial and millennial changes in the mean strength of the South American Summer Monsoon (SASM). In addition,

a composite tree ring chronology from the Altiplano showed that decadal anomalies in SASM-derived precipitation

have been strongly modulated by variability of El Niño-Southern Oscillation (ENSO) over the last 700 year (Morales et

al., 2012). This control is consistent with instrumental measurements which show a significant relationship between

modern ENSO variability and summer precipitation at interannual timescales (Vuille et al., 2000; Knüsel et al., 2005;

Garreaud, 2009). However, it is still unknown if this close relationship extends at timescales longer than decadal.

Evidence from precipitation records covering the last two millennia in the Altiplano are at odds with the modern ENSO-

precipitation relationship (Vuille et al., 2012), providing evidence that this instrumental relationship has not remained

stationary in time. These discrepancies have prompted discussions regarding the existence of different drivers of past

SASM-derived precipitation in the Altiplano, such as Northern Hemisphere temperatures (Vuille et al., 2012), the

latitude of the Intertropical Convergence Zone (ITCZ) (Bird et al., 2011a; Bird et al., 2011b), solar variability (Novello

et al., 2016) and moisture sources in the extra-tropical Atlantic region (Baker et al., 2005; Apaéstegui et al., 2018).

Here we present a new pollen-based precipitation record for the Altiplano that covers the interval between 4500-1000

cal yr BP. The timing and direction (positive/negative) of precipitation anomalies inferred from the pollen data is

evaluated with the primary aim of explore past drivers of precipitation change and assess the evolution of

teleconnections between the Altiplano and other regions in South America, the Pacific and Atlantic Ocean. Our pollen

data were therefore compared with proxy-base records from the Altiplano and the tropical Andes, along with

reconstructions of past ITCZ variability and ENSO-like changes.

## 2 Geographic setting

### 2.1 The Altiplano

The Altiplano (13-22° S) is a high-Andean tectonic plateau that covers an area of ~190,000 km$^2$ encompassing the

central Andes Cordillera of southern Perú, western Bolivia, and northern Chile and Argentina (Fig. 1). Sitting in a

north-south axis over Cenozoic ignimbrites that overlay Cretaceous marine deposits, this high-Andean plateau has

widths ranging from 200 to 300 km and mean elevations between 3000-5000 masl (meter above sea level) (De Silva,

1989; Garzione et al., 2006). Multiple phases of tectonic uplifting and crustal shortening during the last 30 Ma account

for the unusual high elevations of this continental plateau (Isacks, 1988). It has been estimated that the present-day

Altiplano altitude was attained around 6-5 Ma (Garzione et al., 2017). Since that time, continuous internal drainage and

erosional processes has produced a general flattening of the elevated surface resulting in several large-scale plane

fluvio-lacustrine basins (Garzione et al., 2006). Although the Altiplano has been volcanically active since its initial

formation, present magmatism is expressed exclusively in its western margin in the form of a prominent Cretaceous-

Neogene magmatic arc featuring several 5000-6000 masl stratovolcanoes (Allmendinger et al., 1997).

The climate of the Altiplano is cold and semiarid with mean annual temperature and precipitation ranging between 4

and 17° C and 20 to 800 mm, respectively. The pronounced climate gradients observed in stations across this region

result from its large latitudinal extension including subtropical and extra-tropical regions, and altitudinal differences

between the central plateau and its eastern and western flanks. Summer (DJFM) precipitation, both as rain or snow, represents around 70-90 % of the total annual amount (Fig. 2a), and comes in the form of convective storms associated with the development of the SASM in the interior of the continent (Garreaud et al., 2003) (Fig. 2b). Seasonal and interannual precipitation variability in the Altiplano is directly associated with the strength of the SASM, which is in

turn controlled both by the latitudinal position of the ITCZ at the lower atmospheric level (>850 hPa), and by the Bolivian high pressure cell at the upper-level (<300 hPa; Garreaud et al., 2003). In general, above-mean summer precipitation is associated with a southward extension of the ITCZ and a southward position of the Bolivian high pressure cell, which results in an easterly wind anomaly and an strengthening of the SASM (Vuille, 1999; Aceituno and Montecinos, 1993; Garreaud et al., 2009). Easterly SASM moisture comes in two distinctive modes: a northerly

mode originated in the Amazon basin; and a southerly mode sourced in the Gran Chaco basin and the South Atlantic Ocean (Chaves and Nobre, 2004; Vuille and Keimig, 2004). In addition, there is a general consensus that a significant amount of interannual variability in summer precipitation over the Altiplano is influenced by El Niño Southern Oscillation (Garreaud and Aceituno, 2001; Garreaud et al., 2003; Vuille et al., 2000; Sulca et al., 2018). During la Niña phases, the Bolivian high pressure cells sits over the Altiplano, weakening the prevailing westerly flow and

promoting the incursion of convective storms associated with the SASM from the northeast. As a result, above-mean precipitation occurs more often during La Niña years, whereas opposite anomalies tend to occurs during El Niño phases (Fig. 2c).

The vegetation of the Altiplano is rich and diverse, presenting a great number of shrubland, grassland, peatland and forest communities. The distribution of these communities varies locally according to elevation, soil, erosion, water

availability and drainage; and regionally by the gradients in precipitation and temperature described above (Brush, 1982). For the sake of the pollen-climate interpretations of this study, and considering the outstanding diversity of the Altiplano vegetation, we restricted our characterization to the vegetation belts established across the western Andean slopes and the southern Altiplano around our study site. Our brief and simplified description of the regional vegetation communities is based on the much more comprehensive work of Villagrán et al. (1983), Arroyo et al. (1988), Rundel

and Palma (2000), and Orellana et al. (2013). We followed the taxonomic nomenclature adopted by Rodriguez et al. (2018).

The Prepuna vegetation belt established above the upper margin of the absolute desert from ~2000 to 3200 masl, and it is a low density vegetation community (coverage >40%) composed by a floristically diverse assemble of shrubs and herbs. It is dominated by shrubs such as *Ambrosia artemisioides* (Asteraceae), *Atriplex imbricata* (Chenopodiaceae),

and *Aloysia deserticola* (Verbenaceae), with several herbs such as *Descurainia pimpinellifolia* (Brassicaceae), *Balbisia microphylla* (Francoaceae) and *Bryantiella glutinosa* (Polemoniaceae), and scattered cacti such as *Browningia candelaris* and *Corryocactus brevistylus*.





The Puna belt sits above the aforementioned vegetation formation, roughly between 3000 and 4000 masl. It is a floristically more diverse and a more densely vegetated shrubland-grassland community. The Puna belt is dominated by

*Fabiana densa* (Solanaceae), *Baccharis boliviensis* (Asteraceae), *Diplostephium meyenii* (Asteraceae)*, Lophopappus tarapacanus* (Asteraceae) and *Ephedra americana* (Ephedraceae). Other plant natural to the Puna belt are annual herbs such as *Chersodoma jodopappa* (Asteraceae), *Junellia seriphioides* (Verbenaceae), the catus *Cumulopuntia echinacea* (Cactaceae) and the fern *Cheilanthes pruinata* (Pteridophyta). Grasses representatives of the Poaceae family are *Nassella pubiflor, Eragrostis kuschelii.*

A gradual transition from a shrubland- to a grassland-dominated community occurs at the upper limit of the Puna belt. Above 4000 masl, high-Andean steppe tends to replace Puna scrubs, forming a grassland belt characterized by several species of Poaceae family such as *Festuca arundinacea*, *F. chrysophylla*, *Deyeuxia breviaristata*, *Poa lepidula* and several species of the *Anatherostipa* genus. This high elevation steppe also features scattered shrubs such as *Parastrephia lucida* (Asteraceae) and *P. quadrangularis.* Woodlands of the tree *Polylepis tarapacana* (Rosaceae) occur

preferably on the north-facing slopes at elevation above 4100 masl, usually accompanied by species such as *Azorella compacta* (Apiaceae), *Parastrephia teretiuscula* (Asteraceae) and *Baccharis tola* (Asteraceae). The altitudinal limit of vegetation, which sometimes surpasses 5000 masl (Orellana et al., 2013), is occupied by a low shrubland community mostly made of cushion-plants such as *Pycnophyllum molle* (Caryophyllaceae), the cactus *Cumulopuntia ignescens*, and several species of the genera *Werneria* (Asteraceae) and *Nototriche* (Malvaceae), with this latter plant genus usually

found on the characteristic rocky substrates of the mountain slopes.

**2.2 Lago Chungará**

Lago Chungará (18.24° S; 69.15° W; 4570 masl; 23 km$^2$; 40 m maximum depth) is located at the base of Volcán Parinacota on the western border of the southern Altiplano plateau (Fig. 1). The lake sits over Miocene and Pleistocene alluvial, fluvial, lacustrine and volcanic sediments of the Lauca Basin (Gaupp et al., 1999). To the west, the western

Andean flanks connect the basin with the lowlands of the Atacama Desert through a series of river basins and dried canyons (Fig. 1). Lago Chungará was formed by the damming of the Lauca River caused by a debris avalanche during a partial collapse of the ancient Volcán Parinacota caldera, previously dated between 17,000-15,000 cal yr BP (Hora et al., 2007). A recent study using $^{10}$Be surface exposure dates indicate that the collapse of the old cone of Volcán Parinacota led to a major lake expansion at about 8800 cal yr BP (Jicha et al., 2015). The main inlet of the lake is the

Río Lauca to the south and the ephemeral streams Ajata and Sopocalane to the west. Lago Chungará has not surface outlets, although groundwater outflow has been reported (Herrera et al., 2006). The absence of lacustrine deposits above the lake suggests that its present-day water levels are the highest in their history (Herrera et al., 2006). The Lago Chungará area receives between 50-670 mm of precipitation annually (mean 1961-2016 =353 mm), with 88% of this

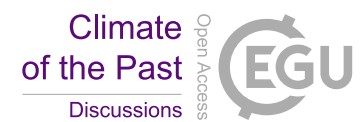



the total amount being received during summer months (DJFM). Mean annual temperature is 4.5° C. The precipitation

at the Lago Chungará heavily contrasts with the Andean slopes and the lowlands of the Atacama Desert to the west, where precipitation plummets to zero below 2500 masl. The close link between ENSO and summer rainfall in the Altiplano mentioned above is clearly seen at Lago Chungará, where instrumental time series shows that around 35-40% of summer rainfall variability (1961-2016) results from variations in the tropical Pacific climatological state, with significant correlation between instrumental rainfall and ENSO (El Niño 3.4 index; Pearson correlation coefficient r = -

0.46; p-value <0.01). The vegetation around the lake is dominated by high-Andean tussock grasses with scattered *Polylepis* woodland.

**2.3 Previous investigations in Lago Chungará**

Lago Chungará has been the focus of several paleolimnological and paleoclimatological studies over the last 20 years. An initial detailed seismic profile of the sediments was presented by Valero-Garcés et al. (2000). A 500-yr

reconstruction of hydrological variability around Lago Chungará was developed a few year later based on sedimentological, geochemical and stable isotopes analyses from a 57cm-long sediment core (Valero-Garcés et al., 2003). In 2002, a field campaign retrieved 15 new sediment cores from different areas of the lake and several publications have emerged from analysis on those sediment sequences. Such studies included a detailed lithological correlation of all cores (Sáez et al., 2007); analysis of the physical properties of the sediments such as magnetic

susceptibility, mineral composition, total organic carbon and grey-colour values (Moreno et al., 2007); a climate reconstruction based on the statistical analysis of mineralogical and chemical parameters and a detailed chronological framework (Giralt et al., 2008); oxygen and carbon isotope composition of diatom silica and their relationship with the hydrological evolution of the lake, as well as solar and ENSO forcings (Hernández et al., 2008, 2010, 2011, 2013); and petrology and isotopic composition of carbonate fraction (Pueyo et al., 2011). More recently, Bao et al (2015) presented

a long-term productivity reconstruction based on fossil diatoms composition and geochemical data, assessing the interplay between regional precipitation, lake levels and organic activity in Lago Chungará. Overall, these studies have provided a detailed paleolimnological and paleoenvironmental history of Lago Chungará that extend for at least 12,800 years, showing a sedimentation regime marked by diatom-rich deposits, interbedded with multiple tephra and carbonate-rich layers (Pueyo et al., 2011). Volcanic input, more frequently recorded after 7800 cal yr BP, was most

likely composed by tephra deposits erupted from Volcán Parinacota (Sáez et al., 2007). Relative high lake levels are recorded before 10,000 cal yr BP, in overall agreement with a pluvial interval documented elsewhere in the Altiplano (Giralt et al., 2008). Dry conditions seem to have prevailed between 8000-3500 cal yr BP (Pueyo et al., 2011), with peak aridity probably occurring between 8000-6000 cal yr BP (Moreno et al., 2007; Giralt et al., 2008; Pueyo et al., 2011). Relative high lake levels prevailed over the last 5000 years (Sáez et al., 2007), superimposed to several high-

amplitude lake stand oscillations after 4000 cal yr BP (Sáez et al., 2007; Giralt et al., 2008). Humid and high lake

stands have been recognized between 12,400-10,000 and 9600-7400 cal yr BP, while dry intervals prevailed between 10,000-9600 and 7400-3500 cal yr BP (Bao et al., 2015). The last 200 years are marked by overall dry conditions (Valero-Garcés et al., 2003). Despite all this information, the climate history around Lago Chungará and its regional drivers over the most recent millennia has not yet been addressed in detail yet (Hernández et al., 2010; Bao et al.,

185    2015).

**3 Methods**

In November 2002, fifteen sediment cores up to 8 m long were obtained from different parts of the lake using a raft equipped with a Kullenberg corer (Sáez et al., 2007). The fossil pollen sequence presented here was developed from Core 7, which was obtained from the northwestern border of the lake at 18 m of water depth (Sáez et al., 2007). In

addition, we obtained pollen data from 26 surface samples collected along west-east transect on the western Andes slope at 18° S with the aim to determine the pollen signals associated with modern vegetation belts. Surface pollen samples were obtained from exposed soils located at regular intervals of ~100 m, from 2100 to 4400 masl. Thus, a surface pollen transect was formed crossing all vegetation belts established across the western Andean slopes up to the base of Lago Chungará (Fig. 1). Furthermore, we obtained annual temperature and summer precipitation data for each

surface sample point from the WorldClim2 dataset (Fick and Hijmans, 2017), with the aim of documenting the climate gradients associated with the pollen transect. All pollen and climate data was used as supporting our paleoclimate inferences.

The 26 surface samples plus 49 fossil pollen samples from Core 7 were prepared following standard procedures for palynological analysis (Faegri and Iversen, 1989) in the Laboratory of Paleoecology and Paleoclimatology of CEAZA,

La Serena, Chile. Pollen data is expressed as relative percentage, which was calculated from the sum of a minimum of 300 terrestrial pollen grains. The percentage of aquatic pollen was calculated from a sum that included all terrestrial pollen and aquatic pollen. Pollen accumulation rate (PAR; particles $yr^{-1}$ $cm^{-2}$) was calculated for terrestrial and aquatic pollen using the data from the age-depth model published by Giralt et al., (2008). Statistical analysis on pollen data included a Stratigraphically Constrained Cluster Analysis (CONISS) for the identification of major pollen changes

(Grimm, 1987).

**4 Results**

**4.1 Stratigraphy and Chronology**

Core 7 comprises sections 7A-1K-1 (146 cm), 7A-2K-1 (151 cm) and 7A-3K-1 (51 cm) with a total length of 348 cm. These core sections correspond to Subunit 2b described in Sáez et al. (2007), composed of dark grey diatomite with

abundant macrophyte and four black tephra layers. These layers are made of andesitic and rhyolitic material with

presence of amphibole (Moreno et al., 2007). The chronological framework used in this study is based in the one

presented in Giralt et al. (2008). The chronology of Core 7 is constrained by 3 AMS radiocarbon dates obtained from

Subunit 2b in cores 11 and 14, which were translated into Core 7 after detailed correlation based on seismic profiles and

tephra keybeds identified as peaks in magnetic susceptibility (Sáez et al., 2007). This chronology uses a modern

reservoir effect of $^{14}$C 3260 years which is assumed to be constant throughout the whole Subunit 2b. A constant

reservoir effect for this unit is supported by multiple lines of evidence, and suggests that the average characteristics of

the lake -such as depth or water volume- did not experience significant changes during the time represented by this

Subunit 2b (Giralt et al., 2008). The chronological framework indicates that Core 7 encompasses the period between

4500-1000 cal yr BP, with depositional rates ranging from 0.26 to 1.31 mm/yr. Unfortunately, the last 1000 years of

sediment history were not recovered during the field campaign.

### 4.2 Modern pollen transect

Our modern surface pollen transect reveals important changes in the abundance of pollen type across Prepuna, Puna and

High Andean Steppe vegetation belts (Fig. 3). Between ~2000 and 3200 masl Prepuna pollen samples are largely

dominated by Chenopodiaceae, Asteraceae (Ast.) *Ambrosia*-type, Brassicaceae *Sisymbrium*-type, B. *Draba*-type and

Ast. Ophryosporus-type. Other pollen types with minor abundances are Portulacaceae *Cistanthe*-type, Juncaceae and

Malvaceae. Between ~3200 and 4000 masl Puna dominant taxa include Ast. *Parastrephia*-type and Ast. *Baccharis*-type

along with *Ephedra* spp., Fabaceae *Prosopis*-type, Solanaceae, Ledocarpaceae *Balbisia*-type and Ast. *Chuquiraga*-type.

Above 4000 masl, surface pollen samples at the high-Andean steppe are less diverse and overwhelmingly dominated by

Poaceae with minor contribution of Apiaceae *Mulinum*-type. Mean annual temperatures associated with the transect

show a sustained decreases from 14° C at 2100 masl to less than 3° C at 4400 masl; summer precipitation, on the other

hand, exhibits a continuous increase from less than 40 mm/yr at 2100 masl to more than 280 mm/yr at the upper limit of

our transect.

### 4.3 Pollen record

Five general zones are recognized based on the major changes in pollen percentages (Fig. 4) and PAR (Fig. 5), assisted

by a CONISS ordination. Overall, the record is largely dominated by Poaceae (mean value 55%), followed by Ast.

*Parastrephia*-type (13%), Chenopodiaceae (8%), Ast. *Baccharis*-type (7%), Apiaceae (6%), Ast. *Ambrosia*-type (4%)

and *Polylepis* spp. (3%). The record mean abundances of *Botryococcus* spp. and *Potamogeton* spp., the two most

abundant aquatic taxa, correspond to 72% and 2% of the total terrestrial sum respectively.

The basal portion of the record (Zone CHU1; 4500-4100 cal yr BP) features above-mean percentage of Poaceae (67%)

and relative low abundances of all other major taxa, especially *Polylepis* spp. which shows minimal presence in this

zone (1%). The PAR record indicates overall very low pollen accumulation for all taxa during this zone.

The following interval (Zone CHU2; 4100-3300 cal yr BP) exhibits a downturn of Poaceae (52%), a rapid and sustained decline in *Botryococcus* spp. (56%), and a significant increase in Ast. *Baccharis*-type (19%) and Ast. *Ophryosporus*-type (2%). These latter two increments are also displayed in the PAR diagram which reveal otherwise relatively low value of all the remaining taxa.

Zone CHU3 (3300-2300 cal yr BP) is characterized by a recovery of Poaceae (59%) and a notable decline in Ast. *Baccharis*-type (3%). Other significant changes in this zone include a rise in *Polylepis* spp. (4%) and sustained increments of Ast. *Parastrephia*-type (17%) and *Botryococcus* spp. (67%). The PAR diagram shows very low values of all terrestrial and aquatic taxa during this entire zone.

Zone CHU4 (2300-1600 cal yr BP) features below-mean abundances of Poaceae (49%) and Ast. *Parastrephia*-type (11%) and marked increments of Apiacea (9%) and *Botryococcus* spp. (81%). This zone also features the appearance of Poaceae *Stipa*-type and P. *Bromus*-type; along with minor peaks in Solanaceae and Fabaceae at the base and the top of this zone respectively. The PAR diagram exhibits a notable increase in all terrestrial taxa and *Botryococcus* spp.

Finally, Zone CHU5 (1600-1000 cal yr BP) exhibits a sustained recovery of Poaceae (56%), a rapid decline in Apiaceae (3%) and a drastic drop of about 50% in *Botryococcus* spp. (72%) in the upper part of the zone. Other notable signatures of this zone are the minor presence of Brassicaceae, above-mean abundance of Ast. *Ambrosia*-type (5%), below-mean Ast. *Baccharis*-type (3%), and rapid increases of Ast. *Ophryosporus*-type and the aquatic *Potamogeton* spp. (3%). The PAR of all terrestrial taxa (excepting Brassicaceae) and *Botryococcus* spp. plummet to near-zero values whereas *Potamogeton* spp. PAR shows noticeable increases.

## 5 Discussion

### 5.1 Pollen-climate relationships

At present, Lago Chungará is situated within the high-Andean steppe vegetation belt and, consistent with this position, pollen assemblages are primarily composed of high-Andean pollen types. This composition suggests that high-elevation steppe vegetation dominated the surroundings of Lago Chungará between 4500-1000 cal yr BP. Nonetheless, the fossil pollen sequence shows a significant representation of pollen types of vegetation belts situated at lower elevations on the western Andes slopes (i.e. Prepuna and Puna belts). For instance, high Andean pollen types (e.g. *Polylepis* spp. [record mean = 3%]) are sometimes presented in lower abundances than Puna or Prepuna elements (e.g. Chenopodiaceae [8%]), although this might also be related to differences in specific pollen production and long distance transport. A considerable altitudinal thermal gradient of up to 10° C is observed in our surface transect which corresponds to a lapse rate of -4.9° C per 1000 m (Fig. 3), a value within the ranges calculated from the Andes of northern Chile and the Altiplano (e.g. Kull et al., 2002; Gonfiantini et al., 2001). Nonetheless, recent low latitude (30° N to 30° S) temperature





reconstructions covering the interval between 7000-100 cal yr BP estimate thermal oscillations of less than 0.8° C in magnitude (Marcott et al., 2013), a value clearly insufficient to explain the changes in vegetation revealed by the Chungará record. Hence, we assume that precipitation rather than temperature variability is the largely responsible for

the vegetation shifts recorded at Lago Chungará. Our surface pollen transect indicates a continuous increase in summer precipitation in the western Andean slopes from 2100 to 4400 masl (Fig. 3), a trend that is well supported by the instrumental record and the previous literature (Villagrán et al., 1981; Latorre et al., 2006). Based on this information, we infer above-mean abundances of high-Andean taxa as humid conditions, and above-mean abundances of Puna and Prepuna pollen as suggestive of dry periods (Latorre et al., 2006; Maldonado et al., 2005; De Porras et al., 2017).

Terrestrial pollen accumulation reflects plant productivity and vegetal coverage, both increasing significantly from the Prepuna belt to the high-Andean steppe. Therefore, terrestrial PAR is used as a proxy for regional humidity. In other words, we interpret pollen variations at Lago Chungará as expansions/contractions of Andean vegetation belts and plant productivity in response to changes in regional summer precipitation. On the other hand, aquatic taxa exhibit notable variations at multiple parts of the sequence rather than a unique monotonic trend (Fig. 4). These multiple variations

suggest a response to underlying climate variations rather than sustained changes in the lake surface area or water depth due to the continuous sediment infilling of the base (Hernandez et al., 2008; Bao et al., 2015). Thus, we interpret variations in aquatic taxa as changes in lake levels driven by regional precipitation (Jankovská and Komárek, 2000). More precisely, below-mean abundances of the freshwater algae *Botryococcus* spp. are interpreted as reduced water levels (Grosjean et al., 2001; Sáez et al., 2007); whereas above-mean abundances of the macrophyte *Potamogeton* spp.,

which is currently presented in shallow waters in the periphery of the lake (Mühlhauser et al., 1995), are interpreted as responding to centripetal expansions of palustral environments due to shallower lake levels (Graf, 1994). Finally, considering that 80-90% of modern precipitation at the Lago Chungará area fall during summer months associated with the development of the SASM (Vuille, 1999; Fig. 2a), we interpret our precipitation changes as responding to long-term variations in the strength of the SASM.

**5.2 Past precipitation trends**

Relative humid conditions are inferred prior to 4100 cal yr BP based on the dominance of Poaceae along with below-mean abundances of all Puna and Prepuna taxa. Above-mean abundances of *Botryococcus* spp. are coherent with this interpretation, suggesting relative low variability in water table and overall increased lake stands. Although Poaceae PAR values are relatively high, total terrestrial PAR is below the long-term mean during this interval, suggesting sparse

vegetation cover. This early period of relative high moisture and stable lake levels occurred during an interval of regional aridity and reduced water levels at Lago Chungará between 9000-4000 cal yr BP (Giralt et al., 2008; Bao et al., 2015), although these studies have shown that this interval was not homogenous, including significant wet/dry variability at sub-millennial timescales.





Between 4100-3300 cal yr BP, the record exhibits lower than mean Poaceae and *Botryococcus* spp.; and above-mean

Ast. *Baccharis*-type and *Ophryosporus*-type. Corresponding PAR increases reveals that the observed changes in the

percentage diagram reflect genuine increments in the abundance of the aforementioned pollen types. Our modern pollen

transect shows that comparable percentages of Ast. *Baccharis* are found below 4000 masl, and therefore their expansion

at Chungará points to a decrease in effective precipitation not lower than 60 mm/year relative to modern values. This

evidence and the very low terrestrial PAR observed during this zone suggest that those few Puna and Prepuna pollen

grains deposited on the lake were most likely windblown from far-distant shrubland communities downslope.

Consistent with a relative dry climate, a drastic drop of *Botryococcus* spp. suggests an overall reduction in lake levels

during this interval.

These changes are followed by a decline in Ast. *Baccharis*-type by 3600 cal yr BP and a sustained expansion of Ast.

*Parastrephia*-type between 3400-2400 cal yr BP; and culminates with a period of persistent above-mean percentages of

Apiaceae between 2400-1600 cal yr BP. The modern distribution of vegetation on the western Andes slopes and our

modern pollen transect shows that Apiaceae is one of the only families which distribution occurred above 4000 a masl

(Fig. 3). In fact, the species *Azorella compacta* is one of the dominant members above 4600 masl, in the altitudinal limit

of vegetation in the Chungará area (Orellana et al., 2013). Hence, the rapid increase of Apiaceae between 2400-1600 cal

yr BP represents a downward expansion of high-Andean and Subnival vegetation. Interestingly, the PAR diagram

shows a rapid expansion of representatives of all vegetation belts between 2400-1800 cal yr BP, expressed as an abrupt

increment in the total terrestrial PAR (Fig. 5). These trends indicates a conspicuous increase of terrestrial plant

productivity that was not restricted to the high-Andean vegetation as suggested by the percentage diagram, but rather to

all vegetation belts downwards. Alternatively, the expansion of cold-tolerant Apiaceae species could have resulted from

a cooling event. However, such a thermal excursion should have been associated with a decrease in plant productivity

around the lake and downwards. Thus, the observed sequence of pollen changes points to a significant increase in

terrestrial plant coverage over the western Altiplano and adjacent Andes slopes, a response that we attribute to a notably

rise in regional precipitation between 2400-1600 cal yr BP. Consistent with this interpretation, both percentage and

PAR of *Botryococcus* spp. show above-mean values during this interval, suggesting persistent high lake stands.

Similarly, diatom assemblages and geochemical data from Lago Chungará agrees with these pollen-climate inferences,

pointing to increased moisture after 3500 cal yr BP (Bao et al., 2015; Giralt et al., 2008; Pueyo et al., 2011), and

overall higher lake levels based on the reduction of periphylic diatoms (Bao et al., 2015) .

The climate trend described above finished abruptly with both the decline in percentage of Apiaceae and Ast.

*Baccharis*-type; and expansions of Brassicaceae, Ast. *Ambrosia*-type and Ast. *Ophryosporus*-type. These latter taxa

correspond to elements commonly present in the lower Prepuna and Puna belts, and therefore their expansion clearly

points toward drier climates between 1600-1000 cal yr BP. The PAR of all major terrestrial taxa shows rapid declines



after 1800 cal yr BP, except for the low-elevation Ast. *Ophryosporus*-type and Brassicaceae. These series of abrupt

changes are suggestive of a general decrease in terrestrial plant productivity in the Altiplano and western slopes which

more likely resulted from a regional decrease in precipitation. This climate interpretation is further supported by the

increase in the PAR of the macrophyte *Potamogeton* spp., suggesting a drastic reduction of lake levels; and by the

relative low lake levels inferred from minor peaks in benthic diatoms at 2200 and 1500 cal yr BP (Bao et al., 2015).

**5.3 Large-scale drivers and teleconnections**

Based on the modern pollen-climate relationships in the Chungará area expose in Sect. 5.1, we interpret the

aforementioned precipitation anomalies as responding to long-term variations in the mean strength of the SASM, with a

weakened SASM between 4100-3300 and 1600-1000 cal yr BP and a notably strengthening between 2400-1600 cal yr

BP. To identify potential drivers of past SASM-related precipitation anomalies at Lago Chungará, Figure 6 shows our

pollen-climate indicators (Fig. 6a-b) and a previously published hydrological reconstructions from Lago Chungará

(Giralt et al., 2008) (Fig. 6c), along with a selection of paleoclimate reconstructions from the Altiplano, the Tropical

Andes and the tropical Pacific region (10° N-18° S). From this composite plot it turns out clear that the inferred dry

conditions for the 4100-3400 cal yr BP interval based on the pollen data seems at odds with the previous reconstruction

of water availability based on geochemical data from Lago Chungará, which exhibits an alternation of dry and humid

conditions without a consistent trend during this interval (Fig. 6c). These differences could be explained in the light of

dissimilar climate-sensitivity of the sediment geochemistry and pollen proxies. Nonetheless, we note that the transition

from humid (2400-1600 cal yr BP) to dry conditions (1600-1000 cal yr BP) revealed by our pollen record agrees with

similar changes in the geochemical data, as well as with a dry event at 1500 cal yr BP detected in the Lago Chungará

diatom profile (Bao et al., 2015; not shown). We will therefore focus our hemispheric comparison to identify the

forcing mechanism behind these two latter climate anomalies.

The centennial wet/dry anomalies experienced in Lago Chungará between 2400-1000 cal yr BP can be reconciled with

some hydrological records from the western Andes slopes and adjacent Atacama Desert. For instance, records based on

plant remains preserved in fossil rodent middens in Salar del Huasco (20° S) reveal the onset of a humid period at 2100

cal yr BP (Maldonado and Uribe, 2012); while [14]C dating of organic deposits at Quebrada Maní (21° S) indicates a

moist interval between 2500-2000 cal yr BP (Gayo et al., 2012). Our Lago Chungará indicators are also clearly

correlated with changes in the δ[18]O values in the Sajama ice core just 35 km to the northeast of Lago Chungará (18° S;

Thompson et al., 1998) (Fig. 6d). This isotopic record exhibits a sustained trend towards isotopically enriched (positive)

values between 2500-1600 cal yr BP, followed by an abrupt depletion (negative trend) between 1600-1200 cal yr BP

(Fig. 6d). The δ[18]O record of the Huascarán ice core (9° S), located more than 1000 km north from Lago Chungará (Fig.

1), does not show a long-term enrichment comparable to the Sajama core (Thompson et al., 1995), although it does





shows a comparable rapid depletion at ~1600 cal yr BP. These overall similarities indicate a coherent regional centennial climate signal in the Altiplano and the Atacama Desert, along with a possible latitudinal gradient where the stronger changes are observed in paleoclimate records south from 18° S.

A comparison between isotopic values in modern rainfall, instrumental time series and proxy data suggests that $\delta^{18}$O values of ice cores from tropical South America reflect primary changes in isotopic water associated with the intensity of the SASM, with depleted values associated with increased SASM intensity and vice versa (Vuille and Werner, 2005). This relationship was used more recently to infer changes in SASM-derived precipitation based on $\delta^{18}$O values from calcite in sediments from Laguna Pumacocha in the Tropical Andes (5° S; Bird et al., 2011a) (Fig. 6g). However, we

note that the enrichment in the Sajama ice core records (weakening of the SASM, reduced precipitation) between 2500-1600 cal yr BP opposes a distinctive depletion (strengthening of the SASM, increased precipitation) at Laguna Pumacocha during this same interval. Isotopic depletion at this time has also been observed in the Lago Junín $\delta^{18}$O record from the Tropical Andes (Fig. 1; Seltzer et al., 2000), which suggests that the isotopic enrichment observed in the aforementioned Sajama and Huascarán ice cores records may not be a direct indicator of changes in SASM activity.

Our Lago Chungará precipitation reconstruction is consistent with this observation, showing a prominent humid anomaly between 2400-1600 cal yr BP which we interpret as a strengthening of the SASM, in line with the Laguna Pumacocha and Lake Junín records from the Tropical Andes. Similarly, the dry interval detected at Lago Chungará between 1600-1000 cal yr BP matches an enrichment trend at Pumacocha, suggesting a regionally coherent pattern of strengthening/weakening of the SASM and associated precipitation anomalies during this time. The climate signal

behind the $\delta^{18}$O variations in ice core records from the Altiplano and the Tropical Andes has been a matter of debate (e.g. Bird et al., 2011a; Thompson et al., 2000). The comparisons presented here suggest that part of the isotopic variability observed in the Sajama and Huascarán ice records between 2600-1000 cal yr BP may not be directly associated with changes in SASM-derived precipitation. We note that precipitation in the Tropical Andes and Altiplano largely occurs during the warmest months, and that a positive relationship between summer precipitation and

temperatures has been documented in the instrumental record (Vuille, 1999). This modern relationship offers an alternative interpretation for the ice core records which, consistent with the other proxies including our Lago Chungará record, suggest that the variability in the Sajama and Huascarán records primary reflex temperature changes, as proposed initially by Thompson et al. (2000), with enriched (depleted) values reflecting increased (decreased) summer temperatures.

The Chungará precipitation anomalies are partially correlated with precipitation records from Lago Titicaca (15° S). A $\delta^{13}$C record from this lake shows a long-term increase in lake levels over the last 4500 years without any appreciable centennial-scale excursion comparable to the observed at Lago Chungará (Baker et al., 2005) (Fig. 6e). In fact, the highest lake levels are recorded between 1400-1000 cal yr BP, coetaneous with the inferred low stands at Lago



Chungará. However, we note that the steady rise in Lago Titicaca water levels halts at ~1400 cal yr BP, suggesting an

interruption in this long-term trend. Notably, a more recent precipitation reconstruction based on δ Deuterium (δD) values on terrestrial leaf waxes from the same lake suggests a marked strengthening of the SASM inferred from rapid isotopic depletion between 2500-1400 cal yr BP (Fornace et al., 2014) (Fig. 6f). Thus, the Titicaca records reveal an overall intensification of the SASM and sustained rise in lake levels until 1400 cal yr BP, and stable lake stands after that time.

In sum, correlations between Lago Chungará and paleoclimate reconstructions from the Atacama Desert, the Altiplano and the Tropical Andes indicates that the centennial-scale precipitation anomalies detected at Lago Chungará were large-scale paleoclimate events caused by strengthening/weakening of the SASM. We further note that the timing and intensity of the Chungará anomalies are more clearly replicated in the southern Altiplano (i.e. Sajama ice core, Lake Titicaca) and Atacama records than in reconstructions closer to the equator (i.e. Laguna Pumacocha; Lago Junín), with

the latter records showing overall less pronounced changes. In fact, the record of metal concentration from the Cariaco Basin off the Venezuelan coast (10° N; Haug et al., 2001) does not show any noticeable trend between 2300-1100 cal yr BP (Fig. 6h), highlighting this latitudinal gradient and suggesting that the centennial-scale SASM changes recorded at Chungará were largely decoupled from north-south shifts of the Intertropical Convergence Zone.

A 2000-yr record of ENSO-like changes based on several published precipitation records in the western and eastern

tropical Pacific Ocean (Yan et al., 2011) offers an interesting comparison with our reconstructed precipitation anomalies. These Pacific records depict predominantly El Niño-like conditions between 2000-1500 cal yr BP, followed by a La Niña-like interval between 1500-950 cal yr BP (Yan et al., 2011) (Fig. 6i). Hence, increased SASM-derived summer precipitation at Lago Chungará occurred predominately under El Niño-like conditions, whereas decreased SASM precipitation occurred under overall La-Niña-like conditions. In fact, the percentage of sands from El Junco

Crater Lake in the Galapagos Island (1° S; Conroy et al., 2008) (Fig. 1), one of the proxies presented in the Yan et al. (2011) ENSO reconstruction, extends the interval of El Niño-like conditions back to 2200 cal yr BP, encompassing the humid anomaly observed at Lago Chungará almost completely. This correlation indicates that the modern ENSO-precipitation relationship in the Altiplano (i.e. high precipitation during La-Niña conditions and vice versa) observed at interannual timescales in the instrumental records (Vuille, 1999), as well as at decadal timescales in tree ring

chronologies (Morales et al., 2012), fails to explain the centennial-scale anomalies detected at Lago Chungará.

Then what climatic mechanisms were driving the observed SASM changes? We note that a latitudinal gradient in precipitation anomalies over the Altiplano and Tropical Andes has already been pointed out in a recent speleothem δ¹⁸O record from the Chiflonkhakha cave system on the eastern slopes of the Andes (18° S; Apaéstegui et al., 2018) (Fig. 1). This study shows that overall humid conditions occurred between 1050-850 cal yr BP, a trend that is progressively



weakening in northern records until is no longer detected in sites north of 10° S. According to the authors, this latitudinal gradient can be explained if the source of precipitation is not tropical in origin, but originated in southeastern South America and the South Atlantic Ocean (>20° S). The lack of correspondence between Lago Chungará and tropical records of past ITCZ and ENSO behaviour, summed to the latitudinal gradient in centennial-scale SASM responses, is consistent with this explanation, suggesting an extra-tropical source of SASM precipitation that could explain the wet/dry anomalies detected in Lago Chungará and other sites in the Atacama Desert-Altiplano region between 2400-1000 cal yr BP. This interpretation is also consistent with the modes of modern Altiplano precipitation presented by Vuille and Kemin (2004), which demonstrated that changes in the source of precipitation can generate distinctive spatial variability across the Altiplano. Whilst present-day inter-annual precipitation in the northern Altiplano (<18° S) and the Tropical Andes is largely tropical in origin and closed linked to modern ENSO and north-south shifts of the ITCZ, convective activity in the southern Altiplano (>18° S) is strongly correlated with humidity fluctuations to the southeast in the Gran Chaco basin (25° S) (Garreaud et al., 2009). Interestingly, a sea surface temperature (SST) reconstruction from Core GeoB6211-1/2 in the South Atlantic coast (32° S) shows increased SST between 2300-1500 cal yr BP and decreased SST between 1500-1100 cal yr BP (Chiessi et al., 2014). In this region, positive SST anomalies are linked to an intensification and northward migration of the South Atlantic Convergence Zone (SACZ), a band of convective activity that connects the South Atlantic to the core of the South American continent (Garreaud et al., 2009) (Fig.2b). The SACZ is a major contributor to the SASM in South Eastern South America, as evidenced in modern climatological studies (Chaves and Nobre, 2004). In fact, a strong connection between the SACZ and the SASM has been suggested in a precipitation reconstruction based on speleothem δ18O variations in Lapa Grande Cave in Southeastern Brazil (14° S; Stríkis et al., 2011) (Fig. 1), which features rapid short-lived increases in precipitation at 2300, 2200 and 1900 cal yr BP, in correspondence with the Lago Chungará positive precipitation anomaly. Thus, based on all this evidence we hypothesize that the precipitation anomalies observed at Lago Chungará and the southern part of Altiplano are explained by centennial-scale changes in the humidity levels to the southeast of South America and the South Atlantic Ocean, which could have been decoupled from changes in the tropics at certain intervals in the past.

**6 Summary and conclusion**

Our late Holocene pollen-based reconstruction from the sedimentary section of Lago Chungará reveals that significant changes in the altitudinal distribution of vegetation communities, terrestrial plant productivity, and lake levels occurred in the Altiplano. We interpret these changes as resulting from anomalies in precipitation associated with changes in the mean strength of the SASM at centennial timescales. In particular we detected dry conditions suggestive of a weakening of the SASM between 4100-3300 and 1600-1000 cal yr BP, and a conspicuous humid interval reflecting an strengthening of the SASM between 2400-1600 cal yr BP. Comparison with multiples paleoclimate records from the

Altiplano and the Tropical Andes shows spatially coherent changes in SASM intensity at centennial timescales which are largely decoupled from records of variations in the position of the ITCZ and ENSO activity. Hence, the close relationship between ENSO and precipitation in the Altiplano documented at interannual and decadal timescales seems

not to be extended to the centennial domain. We further note a clear south-north gradient in the magnitude of precipitation responses, with southern records in the Altiplano and the Atacama Desert responding more markedly than northern sites in the Tropical Andes. All this evidence is consistent with an extra-tropical source of moisture driving centennial-scale changes in SASM precipitation in the former regions, supporting modern climatological studies (Vuille and Keimig, 2004) and interpretations made from recent reconstructions (Apaéstegui et al., 2018). Finally, our results

highlight: (1) the lack of correspondence between past changes in the strength of the SASM at centennial timescales and climate components sourced in the tropics, which are the dominant drivers of the SASM at the present; and (2) a strong teleconnection between the southern Altiplano and the extra-tropics during the most recent millennia. Hence, caution is required in assuming that the tropical drivers of precipitation in the Altiplano represent the exclusive forcings from which future conditions should be expected.

**7 Data Availability**

All data used to interpret the Lago Chungará record is provided in the article. All the information from other records is given in their respective reference on the main text. The pollen data from Lago Chungará will be freely available in the Neotoma paleoecological database (https://www.neotomadb.org/) once this article is accepted for publication. Additional data regarding this investigation can be requested to: Ignacio A. Jara (ignacio.jara@ceaza.cl)

**8 Author contributions**

AS, AH, SG, RB, BVG and AM, carried out the field campaign and obtained the sediments. LG analysed the pollen samples. IAJ and AM designed the study, analysed and interpreted the data. IAJ wrote the manuscript and prepared all the figures with the assistance of AM, AS, AH, SG, RB and BVG.

**9 Competing interests**

The Authors declare that they are no conflict of interest.

**10 Acknowledgments**

This investigation was founded by Fondecyt grant 1181829 and the International Cooperation Grant PI20150081. The Spanish Ministry of Science and Innovation funded this research through the projects ANDESTER (BTE2001-3225), Complementary Action (BTE2001-5257-E), LAVOLTER (CGL2004- 00683/BTE), GEOBILA (CGL2007-

60932/BTE) and CONSOLIDER Ingenio 2010 GRACCIE (CSD2007-00067). In addition, we acknowledge funding



from the Spanish Government through the MEDLANT project. IAJ would like to thanks David López from CEAZA for his assistance in the drawing of Fig. 1-2.

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



**Figure caption**

**Figure 1**. Physiography of the study area in the southern Altiplano (a) Digital Elevation Model (DEM) of the Chungará area depicting Lago Chungará and the position of Core 7, Nevado Sajama and the surface pollen transect, (b) South America DEM with the location of climate reconstructions mentioned in the text (1) Quebrada Maní and Salar del Huasco, (2) Chiflonkhakha cave system, (3) Lago Titicaca, (4) Lago Junín, (5) Nevado Huascarán, (6) Laguna Pumacocha, (7) El Junco Crater Lake, (8) Cariaco Basin, (9) Lapa Grande Cave, (10) Core GeoB6211-1/2.

**Figure 2**. South America atmospheric features based on reanalysis data (a) Summer (DJFM) percentage of the total annual precipitation amount based on the ERA-40 reanalysis dataset (1980-2017), (b) Mean summer (DJFM) winfields (m s$^{-1}$) at 850 hPa for the 1980-2017 period from the ERA-Interim reanalysis, (c) Correlation (1980-2017) between South America summer precipitation (DJFM) and the Southern Oscillation Index (SOI) during the austral summer (DJFM). All precipitation data correspond to the ERA-40 reanalysis dataset which is freely provided by the European Centre for Medium-Range Weather Forecasts (ECMWF). The SOI is freely provided by the National Oceanic and Atmospheric Administration (NOAA), USA. ITCZ = Intertropical Convergence Zone, SACZ = South Atlantic Convergence Zone, SASM = South American Summer Monsoon.

**Figure 3**. Diagram with the elevation and pollen assemblages of the 26 surface samples collected along W-E transect on the western Andes slope at 18° S. The altitudinal distribution of the main vegetation belts of the western Andean slopes and the Altiplano is shown in the right panel along with the summer precipitation and annual temperature obtained with using the WorldClim2 dataset (Fick and Hijmans, 2017).

**Figure 4**. Terrestrial and aquatic pollen percentages from Core 7 at Lago Chungará records plotted against depth and age scales. The diagram show the main zones of the records determined with the aid of a CONISS cluster analysis.

**Figure 5**. Terrestrial and aquatic pollen accumulation rate (PAR, grain cm$^{-2}$ yr$^{-1}$) of the Lago Chungará sediment section. PAR values are calculated using the depositional time obtained from the age-depth model published by Giralt et al. (2008).

**Figure 6**. Summary plot including key Lago Chungará climate indicators (black curves) and selected paleoclimate records from the Tropical Andes and the Pacific Ocean, (a) Apiaceae pollen percentage, (b) *Botryococcus* spp. percentage, (c) Lago Chungará second eigenvectors (15%) of Principal Component Analysis from geochemical dataset (Giralt et al., 2008), (d) δ$^{18}$O record of ice water from Sajama Ice core (Thompson et al., 2000), (e) δ$^{13}$C record from Lago Titicaca (Baker et al., 2005), (f) δD record of terrestrial leaf waxes from Lago Titicaca (Fornace et al., 2014), (g) δ$^{18}$O record of calcite in sediments from Laguna Pumacocha (Bird et al., 2011), (h) %Ti from Cariaco Basin (Haug et al., 2001), (i) Southern Oscillation Index (SOI) reconstruction from eastern and western tropical pacific records (Yan et





al., 2011). Yellow bands indicate dry anomalies, whereas the green band shows the wet anomaly as recorded in Lago

Chungará.



Figure 1

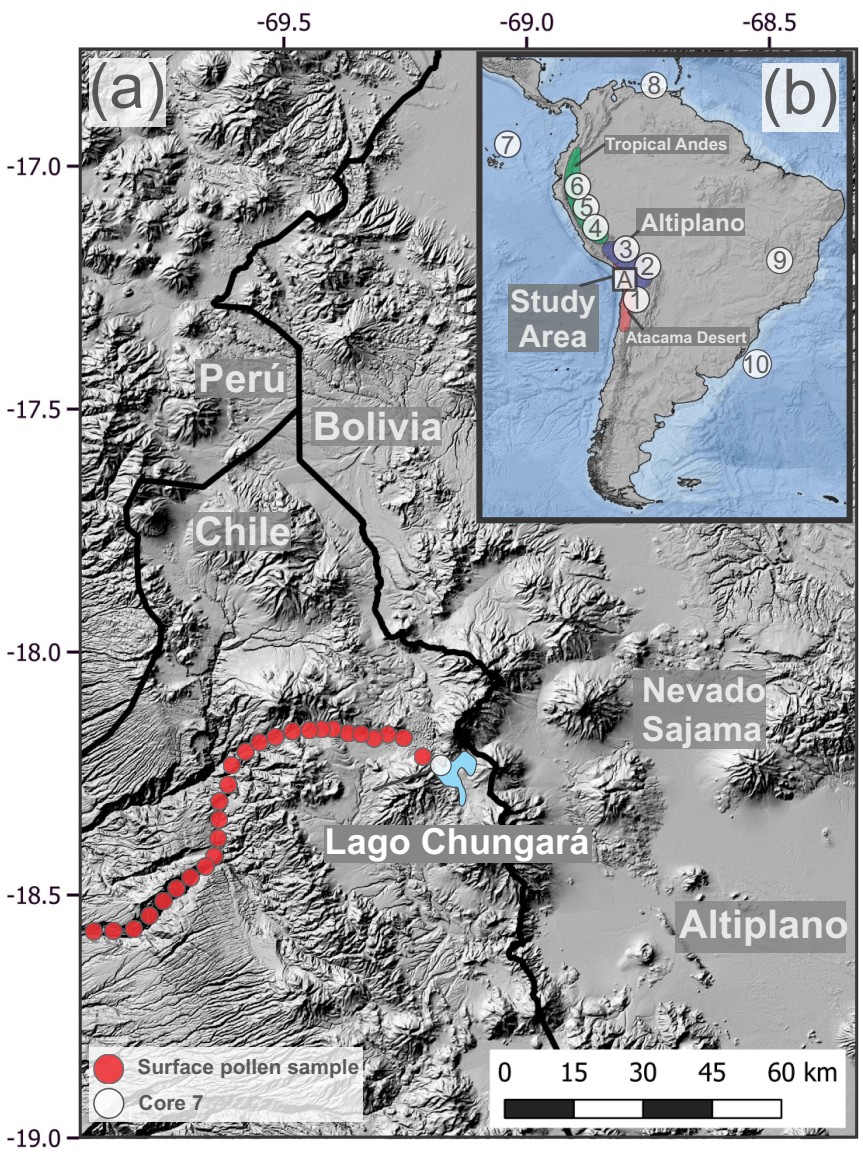



Figure 2

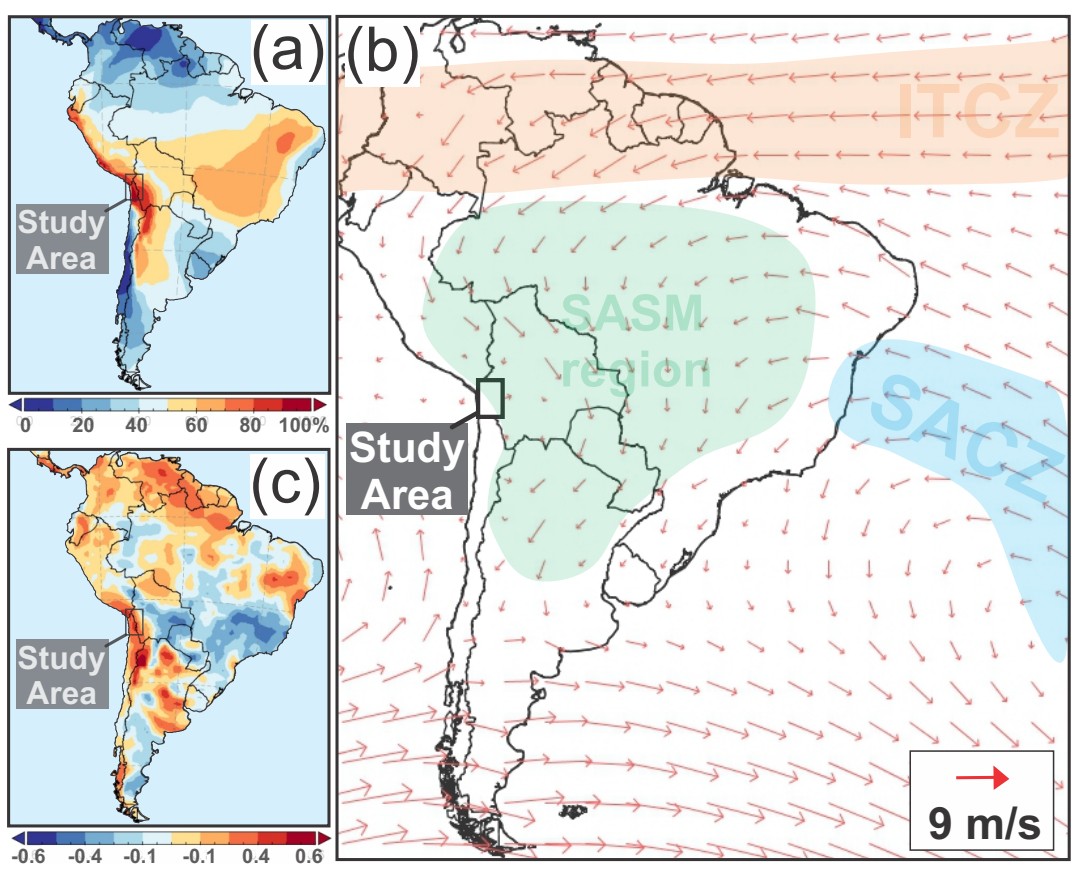



Figure 3

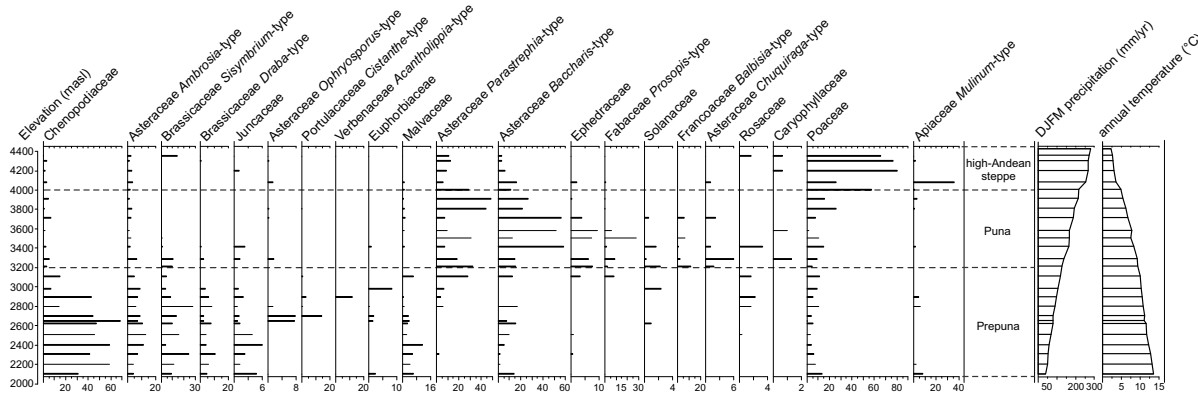





Figure 4

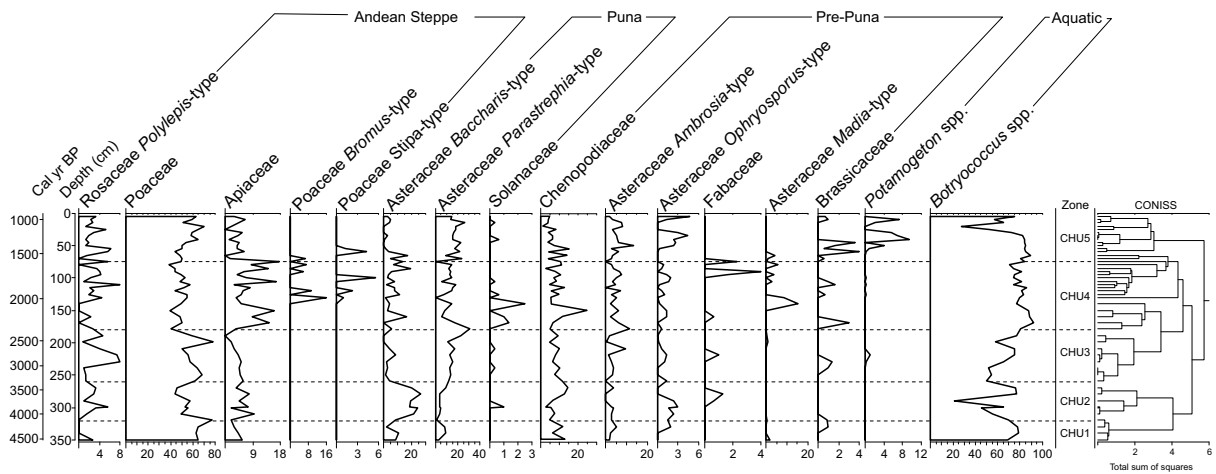





Figure 5

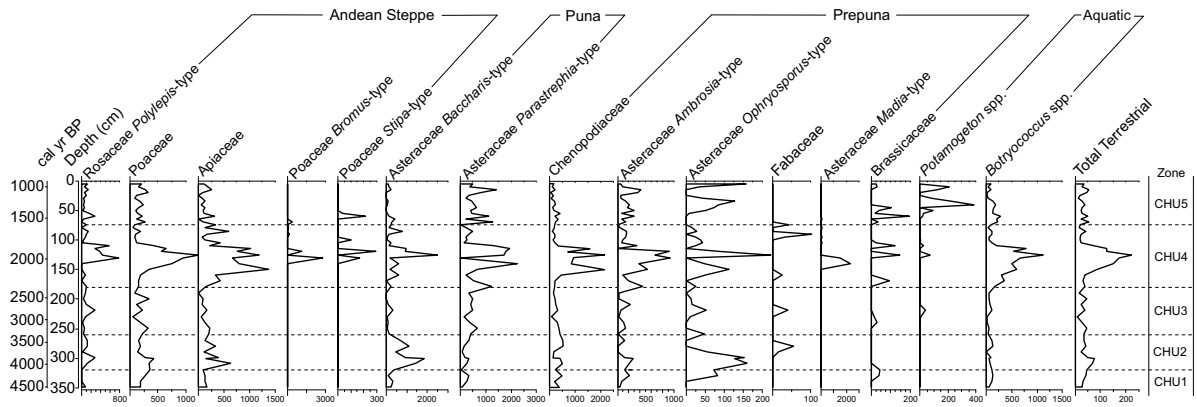



Figure 6

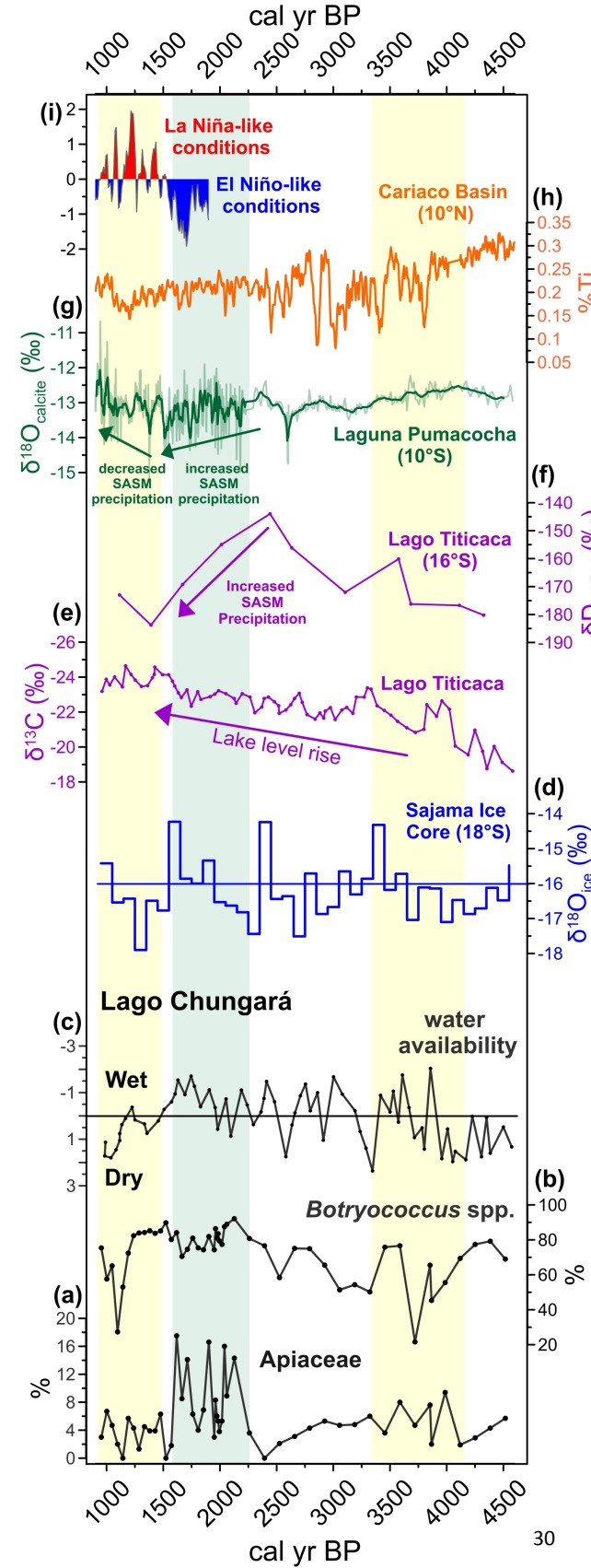