# Peer review of "Centennial-scale precipitation anomalies in the southern Altiplano $(18^{\circ}\ S)$ suggest an extra-tropical driver for the South American Summer Monsoon during the late Holocene"

_Climate of the Past, 2019_

## Referee Comment (RC1) · Anonymous Referee #1 · 20 Mar 2019

This study presents a Holocene pollen record from a lake in the Chilean Altiplano. Overall this is an interesting record that deserves being published, in particular because it stems from a region where our understanding of past changes in climate is still rather rudimentary. I only have a few major comments and a number of smaller edits related to minor grammatical errors.

Main comments:

There is very little discussion (and no Figures) explaining the age-depth relationship

of this sediment record. There appear to have been significant corrections (reservoir effect) applied to the data, but there is little to no supporting information on how exactly the core was dated and how the chronology was established. If I understand correctly, the dating is based on only 3 radiocarbon dates. I understand that the chronology was established and discussed in an earlier paper, but nonetheless, to aid with the interpretation, more information is required to assess the chronological uncertainties in this part of the core.

There are significant uncertainties associated with the age chronology of both Sajama and Huascaran ice core records. Hence I do not have too much faith in the discussion comparing the Chungara record with the data from these two sites (section 5.3). It might be better to compare your record with higher resolution and more accurately (U/Th) dated ice core or speleothem records instead. For example the Holocene record produced by Kanner et al. (2013) from Huagapo cave is located closer to your site than Huascaran, is of much higher resolution, and much better age-constrained. The same is true for the Quelccaya ice core record, although it does not cover the full Holocene. Furthermore, the Huagapo cave record is more consistent with your data, also showing a clear switch from wet conditions at 2 ky BP to much drier conditions peaking around 1.5 ky BP.

While I agree that the main changes affecting the pollen record were driven by precipitation rather than temperature changes over the course of the Holocene, the estimated maximum temperature change in the region (0.8 C) is almost certainly too low. The region has seen significant glacial advances and retreat phases during the Holocene (e.g. Jomelli et al., 2011), most notably associated with the Little Ice Age (LIA) period (e.g. Rabatel et al., 2013). Although the LIA glacial advance in the region may have been partially caused by increased moisture, it is quite clear that the temperature reduction in the region must have been larger than the 0.8 C cited in the paper. Jomelli et al. (2011) based on glacial modeling, for example, estimate a regional LIA cooling of 2.1 C. The reported cooling of 0.8C is an estimate that is based on a large-scale

circum-tropical average and should not be applied to any single location. Data from local glacier reconstructions provide much better, regionally applicable constraints.

Page 13: Again, I would caution regarding the direct comparison of high-resolution, accurately dated records such as the lake record from Pumacocha (annually resolved for the last 2.3 ky years) and ice core records which are poorly resolved and have age uncertainties that are orders of magnitude larger. The disagreement between these records is likely due to chronological uncertainties rather than different climate sensitivities of the isotopic record, which would be very difficult to explain. In fact, over the period where the chronological control is strong, ice core records (Quelccaya), speleothem records (Huagapo, Palestina) and lake records (Pumacocha) are in agreement. Furthermore, evidence from observational calibration studies on Sajama (Hardy et al., 2003) and ice core forward modeling (Hurley et al., 2016) clearly document that Andean ice core d18O is a faithful recorder of precipitation and the South American summer monsoon strength and that the isotopic signal associated with temperature changes is inconsistent with what is being recorded in the ice core record (Hurley et al. 2019).

Minor edits: Line 43: 'period of instrumental' Line 45: 'makes is difficult' Line 66: 'of exploring', 'and assessing' Line 79: 'have produced' Line 95: 'originating' Line 121: 'plants' Line 122: 'cactus' Line 123: 'representative' Line 145: 'has no' Line 179: 'superimposed on' Line 211: 'based on' Line 245: 'values for' Line 280: 'vegetation cover' Line 321: 'trends indicate' Line 326: 'notable rise' Line 344: 'notable strengthening' Line 342: ' discussed in Sect. 5.1' Line 348" replace 'turns out clear' with 'is evident' Line 349" 'seem at odds' Line 369: 'south of' Line 437: 'Keimig' Line 446: 'southeastern' Line 460: 'reflecting a' Line 479: 'requested from' Line 485: 'they have no' Line 487: 'was funded by' Line 689: 'wind fields'

References cited:

Hardy et al., 2003: Variability of snow accumulation and isotopic composition on Nevado Sajama, Bolivia. J. Geophys. Res., 108, D22, 4693, doi: 10.1029/2003JD003623.

Hurley et al., 2016: Forward modeling of $\delta$18O in Andean ice cores. Geophys. Res. Lett., 43(15), 8178-8188, doi:10.1002/2016GL070150.

Hurley et al., 2019: On the interpretation of the ENSO signal embedded in the stable isotopic composition of Quelccaya Ice Cap, Peru. J. Geophys. Res., 124, 131-145, doi:10.1029/2018JD029064.

Jomelli et al., 2011: Irregular tropical glacier retreat over the Holocene driven by progressive warming. Nature, 474, 196-199.

Kanner et al., 2013: High-resolution variability of the South American summer monsoon over the last seven millennia: Insights from a speleothem record from the central Peruvian Andes. Quat. Sci., Rev., 75, 1-10.

Rabatel et al., 2013: Current state of glaciers in the tropical Andes. A multi-century perspective on glacier evolution and climate change. Cryosphere, 7, 81-102.

---

## Referee Comment (RC2) · Anonymous Referee #2 · 22 May 2019

This article presents an interesting pollen record from Lago Chungará and a current pollen distribution along an adjacent transect on the western Andes slope. The authors, based on the pollen record, reconstruct the precipitation variability at centennial scale during the Late Holocene. They compare their results with paleoclimate sites along Atacama Desert, the Altiplano and the Tropical Andes, and suggest an extra-tropical driver for the South American Summer Monsoon (SASM). The contribution is interesting and it deserves to be published. Nevertheless, I have some major comments related with the chronology of the record and with the discussion regarding El

Niño – Southern Oscillation (ENSO) and the Intertropical Convergence Zone (ITCZ).

Major comments:

1) I agree with the referee #1 regarding the chronology. The chronological model should be available in the article, and not hidden in two earlier articles. The article should illustrate, in a figure, the chronology for the Core 7 and its correlation with the Subunit 2b in cores 11 and 14. It also should present a more detailed information to assess the chronological uncertainties. Furthermore, a summary figure with the earlier data cited in the item 2.3 (Previous investigations in Lago Chungará) is missing, as well as a more complete discussion considering the available data (e.g. sedimentation rate, geochemical data, etc)

2) The discussion related to the extra-tropical driver for the SASM is not strong enough. If the chronology of the record is right, the discussion about ENSO decoupling is interesting, but it needs to be more extended (only two references are cited: Yan et al., 2011; Conroy et al., 2008). The discussion would be more robust if the authors consider multiple records that propose the enhanced ENSO during the Late Holocene, specifically the intensification trend of El Niño for the last millennia (Vargas et al., 2006; Zhang et al., 2014; Barr et al., 2019; Ortega et al., 2019). On the other hand, it seems that the discussion to discard the ITCZ influence of SASM during 2400 to 1000 cal yr BP is incomplete. The authors just indicate that "the record of metal concentration from Cariaco Basin off the Venezuela coast (10°N; Haug et al., 2001) does not show any noticeable trend between 2300-1100 cal yr BP" discarding the north-south shifts of the ITCZ in the centennial-scale variability of SASM. I suggest analyzing the ITCZ position by mean of the anomaly of the Titanium (%) from Haugh et al. (2001) as in Salvatecci et al. (2014) (Figure 3 in Salvatecci et al., 2014). Considering this analysis, the authors will note a different trend between the periods 2500-1500 cal yr BP and 1500-1000 cal yr BP, the influence of the ITCZ will be clearer and probably more difficult to remove it completely as a driver of the SASM. I also suggest adding Sach et al. (2018) to the discussion.

Minor comment:

Line 76-77. It is necessary to add updated references on the geological setting. Other authors (Barnes and Ehlers, 2009; Jordan et al., 2010) point out that slow steady-state uplift since 40 Ma is more consistent than enhanced short periods of uplift.

Line 216. "...is supported by multiple lines of evidence". This sentence is too imprecise. References are needed.

Line 355. What kind of geochemical data?. As it is mentioned above, a figure with the available data (such as Bao et al., 2015) is necessary.

Suggested references:

Barnes, J.B., Ehlers, T.A., 2009. End member models for Andean Plateau uplift. Earth-Sci. Rev. 97, 105–132. https://doi.org/10.1016/j.earscirev.2009.08.003

Barr C, Tibby J, Leng MJ, et al. Holocene El Niño-Southern Oscillation variability reflected in subtropical Australian precipitation. Sci Rep. 2019;9(1):1627. Published 2019 Feb 7. doi:10.1038/s41598-019-38626-3

Jordan, T. E., Nester, P.L., Blanco, N., Hoke, G.D.; Dávila, F., Tomlinson, A.J., 2010. Uplift of the Altiplano‐Puna plateau: A view from the west. Tectonics 29, TC5007, doi:10.1029/2010TC002661.

Ortega, C., Vargas, G., Rojas, M., Rutllant, Muñoz, P., Lange, C.B., Pantoja, S., Dezileau, L., Ortlieb, L., 2019. Extreme ENSO-driven torrential rainfalls at the southern edge of the Atacama Desert during the Late Holocene and their projection into the 21th. Century Global and Planetary Change 175, 226–237

Sachs, J. P., Blois, J. L., McGee, T., Wolhowe, M., Haberle, S., Clark, G., & Atahan,P.(2018).Southwardshiftofthe Pacific ITCZ during the Holocene. Paleoceanography and Paleoclimatology, 33, 1383–1395. https://doi.org/10.1029/2018PA003469

Salvatteci, R., Gutiérrez, D., Field, D., Sifeddine, A., Ortlieb, L., Bouloubassi, I., Boussafir, M., Boucher, H., and Cetin, F.: The response of the Peruvian Upwelling Ecosystem to centennial-scale global change during the last two millennia, Clim. Past, 10, 715-731, https://doi.org/10.5194/cp-10-715-2014, 2014.

Vargas, G., Rutllant, J., Ortlieb, L. 2006. ENSO tropical – extratropical climate teleconnections and mechanisms for Holocene debris flows along the hyperarid coast of western South America (17°–24°S). Earth Planet.Sci.Lett.249,467– 483

Zhang, Z., Leduc, G., Sachs, J. P. (2014) El Niño evolution during the Holocene revealed by a biomarker rain gauge in the Galápagos Islands. Earth and Planetary Science Letters, Elsevier, 404, pp.420-434. ff10.1016/j.epsl.2014.07.013ff. ffhal-0147212

---

## Author Comment (AC1) · 10 Jun 2019

Referee #1 provided a complete and very constructive review, including some of the most important topic address in our original manuscript. The Referee raised 4 major points. Here are point-by-point responses to these issues:

1.1. Little discussion of the Chungará chronology

Response: According to Referee #1 more information is required to evaluate the chronology of our record. We partially disagree with this suggestion because -as

indicated in the manuscript (lines 160-162)- the radiocarbon chronology used in our study was explained and discussed in detail in a peer-reviewed publication (Giralt et al., 2008). In fact, this chronology has subsequently been used in multiple peer-reviewed studies from Lago Chungará (e.g. Bao et al., 2015; Hernández et al., 2013) with references to the Giralt et al (2008) article. For that reasons we decided not to include any table or diagram in the main text, although the main features of the chronology are explained throughout lines 202-210. Nonetheless, we agree with Referee #1 that more supporting information might be desirable. Hence, we add the details of the radiocarbon dates, the estimated reservoir correction, and the age-depth curve in the form of a new table and figure in the Supplementary Material (see Figure S1 and Table S1 and attached doc file to this response).

1.2. No reliability in comparing Lago Chungara and Sajama and Huascarán records

Response: Referee #1 comments that a comparison with the Sajama and Huascarán ice records is not appropriated since these records have significant uncertainties in their chronologies. Although we agree with the limitation of these ice records, we note that the Sajama ice cap is the closest paleoclimate site to Lago Chungará, less than 35 km apart from each other (Fig 1 in the original manuscript). Thus, a comparison between these two records is relevant to explore the spatial extent of the climate inferences of our manuscript. In addition, we content that there is reliability in comparing these two records based on stratigraphic evidence. For instance, Giralt et al. (2008) clearly demonstrate that the onset of the Holocene volcanic activity of the Parinacota volcano is recorded at both side at around 7500 cal yr BP (Fig. 8 of that article). Furthermore, there is an excellent correlation between the main eruptions of the Parinacota volcano present in Lago Chungara and the dust peaks recorded in the Sajama ice core for the 4500-1000 cal yr BP period, as it was explained in Sáez et al., (2015). These studies support the idea that both climate reconstructions are perfectly comparable, despite their potential chronological uncertainties. For all these reasons we decided to include a comparison of the $\delta$18O record of the Sajama ice record in the main

text and in Fig. 6. On the other hand, we agree with Referee #1 that a comparison with the Huascarán record-located several thousand kilometres away from Chungará- does not represent a real contribution to our discussion, and therefore we have removed that record from the text and from Fig. 6. Additionally, Referee #1 suggests that it might be better to compare our record with reconstructions depicting higher resolutions more accurate chronologies. We have followed this constructive advice, adding a direct comparison with the $\delta$18O record from the Huagapo cave (lines 392-394 and Fig. 6g). As mentioned by Reviewer #1, this record is consistent with our interpretations from the Chungará record, providing additional support for our paleoclimate inferences.

1.3. Regional temperature variability presented in the manuscript is too low

Response: Referee #1 agrees with us that precipitation is the main driver of the observed pollen changes, although she/he considers that our estimation for the magnitude of Holocene temperature change is too low. We used the values presented in Marcott et al. (2013) for the 4500-1000 cal yr BP period, which are lower than 0.8° C in magnitude for the whole low-latitude band (30° N-30° S). We agree with Referee #1 that this is a broad estimation and that glacier reconstructions from the Tropical Andes provide a more accurate regional constrain that should also be discussed. Consequently, we have added new estimations for the amplitude of Holocene temperature fluctuations in lines 261-266, following the referee's suggested literature. Here we provide a justification of the specific temperature values selected: The Jomelli et al. (2011) article mentioned by Referee #1 uses a climate-glacier model to suggest that temperatures during the Little Ice Age (600-100 cal yr BP) could have been as much as 2.1 ± 0.8°C lower than pre-industrial means, a value considerably higher than our initial suggestion. However, the application of this estimation to the Chungará record should be taken with caution for two main reasons. First, because our record does not cover the LIA interval, and therefore this climate event and associated temperature decline should only be used as referential values. Secondly, this estimation was made using precipitation values corresponding to the early- and mid-Holocene periods (prior to 4000 cal

yr BP), which are considerably lower than late Holocene estimations. Critically, the model does not provide with a unique solution for past temperatures reconstruction, as there are several temperature/precipitation combinations than can reproduce the observed glacial fluctuation. In particular, the magnitude of the LIA cooling declines rapidly if precipitation increases in the model. This is a relevant point because there is evidence from multiple independent records across the Tropical Andes indicating that LIA was a humid interval with precipitation significantly above pre-industrial values (e.g. Vuille et al., 2012). Therefore, we content that -if considering- the value of 2.1 $\pm$ 0.8°C should be taken as a maximum for the magnitude of thermal variability for the Chungará record. Referee #1 also mentioned the more recent review of past glacial fluctuations in the Central Andes presented by Rabatel et al. (2013). Interestingly, the authors of this article also agree that conditions were more likely wetter during the LIA, which could have increased glacier accumulation rates, driving glacial advancement along with colder temperatures. According to these authors, temperature could have drop as much as 3.2 $\pm$ 1.4°C in the Andes of Venezuela (8° N), although this is hardly a local constrain for Lago Chungará. The closest climate-glacial reconstruction cited in Rabatel et al. (2013) corresponds to the records from the Bolivian Andes (Rabatel et al., 2008), which reconstructs a LIA temperature decrease of about 1.1-1.2°C. We consider this could be in fact a better estimation than the one presented in Jomelli et al. (2011), since it assumes a precipitation input 20-30% above the present. Based on all this information, we the range from 1.1 to 2.2° C a reasonable estimation for the magnitude of temperature change during the interval of the Chungará record.

1.4. Inappropriate comparison between well-dated records and ice core

Response: Referee #1 notes that a direct comparison between highly resolved, well-dated records such as Laguna Pumacocha and the less well-dated Sajama and Huascarán ice records is problematic due to the high chronological uncertainties inherent to the ice reconstructions. Refereee #1 further challenge our interpretation that differences between $\delta$18O records from lake and ice cores in the Central Andes could result

from the fact that ice-based reconstructions are probably sensing temperature fluctuations instead of SASM variations. We consider this a valid and constructive suggestion. We also acknowledge that discussing the climate sensitivity of ice core records is beyond the scope of our manuscript and clearly very difficult to explain. Therefore, we have removed the discussion that led to the suggestion that the $\delta18O$ signal in ice core record could be responding to temperatures from our original manuscript in page 12 (lines 370-394). Instead, we have indicated that these differences are more likely resulting from chronological uncertainties based on: (1) a good agreement between lake, speleothem and ice record during period of strong chronological control, and (2) a trustable SASM signal in $\delta18O$ record from ice cores.

Minor edits of Referee #1: Line 43: 'period of instrumental' Response: CORRECTED Line 45: 'makes is difficult' Response: CORRECTED Line 66: 'of exploring', 'and assessing' Response: CORRECTED Line 79: 'have produced' Response: CORRECTED Line 95: 'originating' Response: CORRECTED Line 121: 'plants' Response: CORRECTED Line 122: 'cactus' Response: CORRECTED Line 123: 'representative' = CORRECTED Line 145: 'has no' Response: CORRECTED Line 179: 'superimposed on' Response: CORRECTED Line 211: 'based on' Response: CORRECTED Line 245: 'values for' Response: CORRECTED Line 280: 'vegetation cover' Response: CORRECTED Line 321: 'trends indicate' Response: CORRECTED Line 326: 'notable rise' Response: CORRECTED Line 344: 'notable strengthening' Response: CORRECTED Line 342: ' discussed in Sect. 5.1' Response: CORRECTED Line 348" replace 'turns out clear' with 'is evident' Response: REPLACED Line 349" 'seem at odds' Response: CORRECTED Line 369: 'south of' Response: REMOVED Line 437: 'Keimig' Response: CORRECTED Line 446: 'southeastern' Response: CORRECTED Line 460: 'reflecting a' Response: CORRECTED Line 479: 'requested from' Response: NOT CONSIDERED Line 485: 'they have no' Response: CORRECTED Line487: 'was funded by' Response: CORRECTED Line 689: 'wind fields' Response: CORRECTED

REFERENCES

Bao, R., Hernández, A., Sáez, A., Giralt, S., Prego, R., Pueyo, J., Moreno, A., and Valero-Garcés, B. L.: Climatic and lacustrine morphometric controls of diatom paleoproductivity in a tropical Andean lake, Quaternary Science Reviews, 129, 96-110, 2015.

Barnes, J., and Ehlers, T.: End member models for Andean Plateau uplift, Earth-Science Reviews, 97, 105-132, 2009.

Barr, C., Tibby, J., Leng, M., Tyler, J., Henderson, A., Overpeck, J., Simpson, G., Cole, J., Phipps, S., and Marshall, J.: Holocene el Niño–southern Oscillation variability reflected in subtropical Australian precipitation, Scientific reports, 9, 1627, 2019.

Conroy, J. L., Overpeck, J. T., Cole, J. E., Shanahan, T. M., and Steinitz-Kannan, M.: Holocene changes in eastern tropical Pacific climate inferred from a Galápagos lake sediment record, Quaternary Science Reviews, 27, 1166-1180, http://dx.doi.org/10.1016/j.quascirev.2008.02.015, 2008.

Giralt, S., Moreno, A., Bao, R., Sáez, A., Prego, R., Valero-Garcés, B. L., Pueyo, J. J., González-Sampériz, P., and Taberner, C.: A statistical approach to disentangle environmental forcings in a lacustrine record: the Lago Chungará case (Chilean Altiplano), Journal of Paleolimnology, 40, 195-215, 2008.

Haug, G. H., Hughen, K. A., Sigman, D. M., Peterson, L. C., and Röhl, U.: Southward migration of the intertropical convergence zone through the Holocene, Science, 293, 1304-1308, 2001.

Hernández, A., Bao, R., Giralt, S., Sáez, A., Leng, M. J., Barker, P. A., Kendrick, C. P., and Sloane, H. J.: Climate, catchment runoff and limnological drivers of carbon and oxygen isotope composition of diatom frustules from the central Andean Altiplano during the Lateglacial and Early Holocene, Quaternary Science Reviews, 66, 64-73, 2013.

Jomelli, V., Khodri, M., Favier, V., Brunstein, D., Ledru, M.-P., Wagnon, P., Blard, P.-H.,

Sicart, J.-E., Braucher, R., and Grancher, D.: Irregular tropical glacier retreat over the Holocene epoch driven by progressive warming, Nature, 474, 196, 2011.

Jordan, T. E., Nester, P. L., Blanco, N., Hoke, G. D., Dávila, F., and Tomlinson, A.: Uplift of the Altiplano–Puna plateau: A view from the west, Tectonics, 29, 2010.

Marcott, S. A., Shakun, J. D., Clark, P. U., and Mix, A. C.: A reconstruction of regional and global temperature for the past 11,300 years, science, 339, 1198-1201, 2013.

Ortega, C., Vargas, G., Rojas, M., Rutllant, J. A., Muñoz, P., Lange, C. B., Pantoja, S., Dezileau, L., and Ortlieb, L.: Extreme ENSO-driven torrential rainfalls at the southern edge of the Atacama Desert during the Late Holocene and their projection into the 21th century, Global and Planetary Change, 175, 226-237, 2019.

Rabatel, A., Francou, B., Jomelli, V., Naveau, P., and Grancher, D.: A chronology of the Little Ice Age in the tropical Andes of Bolivia (16 S) and its implications for climate reconstruction, Quaternary Research, 70, 198-212, 2008.

Rabatel, A., Francou, B., Soruco, A., Gomez, J., Cáceres, B., Ceballos, J. L., Basantes, R., Vuille, M., Sicart, J. E., Huggel, C., Scheel, M., Lejeune, Y., Arnaud, Y., Collet, M., Condom, T., Consoli, G., Favier, V., Jomelli,

V., Galarraga, R., Ginot, P., Maisincho, L., Mendoza, J., Ménégoz, M., Ramirez, E., Ribstein, P., Suarez, W., Villacis, M., and Wagnon, P.: Current state of glaciers in the tropical Andes: a multi-century perspective on glacier evolution and climate change, The Cryosphere, 7, 81-102, 10.5194/tc-7-81-2013, 2013.

Sáez, A., Giralt Romeu, S., Hernández Hernández, A., Bao Casal, R., Pueyo Mur, J. J., Moreno Caballud, A., and Valero Garcés, B. L.: Comment on" Climate in the Western Cordillera of the Central Andes over the last 4300 years", by Engel et al.(2014), Quaternary Science Reviews, 2015, vol. 109, p. 126-130, 2015.

Salvatteci, R., Gutiérrez, D., Field, D., Sifeddine, A., Ortlieb, L., Bouloubassi, I., Boussafir, M., Boucher, H., and Cetin, F.: The response of the Peruvian Upwelling Ecosystem to centennial-scale global change during the last two millennia, Climate of the Past, 10, 715-731, 2014.

Vargas, G., Rutllant, J., and Ortlieb, L.: ENSO tropical–extratropical climate teleconnections and mechanisms for Holocene debris flows along the hyperarid coast of western South America (17–24 S), Earth and Planetary Science Letters, 249, 467-483, 2006.

Vuille, M., Burns, S., Taylor, B., Cruz, F., Bird, B., Abbott, M., Kanner, L., Cheng, H., and Novello, V.: A review of the South American monsoon history as recorded in stable isotopic proxies over the past two millennia, Climate of the Past, 8, 1309-1321, 2012.

Zhang, Z., Leduc, G., and Sachs, J. P.: El Niño evolution during the Holocene revealed by a biomarker rain gauge in the Galápagos Islands, Earth and Planetary Science Letters, 404, 420-434, 2014.

Please also note the supplement to this comment:
https://www.clim-past-discuss.net/cp-2019-13/cp-2019-13-AC1-supplement.pdf

[Figure]

**Fig. 1.** Figure S1

| Laboratory code | core | depth (cm) | $^{14}C$ age | 1σ | Median probability (cal yr BP) | youngest 2σ intercept (cal yr BP) | oldest 2σ intercept (cal yr BP) | Calibration curve |
|---|---|---|---|---|---|---|---|---|
| Poz-8726 | 14 A-1 | 100 | 4620 | 40 | 2010 | 1791 | 2188 | SHCal13 |
| Poz-8720 | 11 A-2 | 165 | 4850 | 40 | 2263 | 2073 | 2382 | SHCal13 |
| Poz-8721 | 11 A-2 | 270 | 7290 | 50 | 3468 | 2658 | 4263 | SHCal13 |

**Fig. 2.** Table S1

---

## Author Comment (AC2) · 10 Jun 2019

Referee #2 also discussed key issues and provided constructive comments on the original manuscript. The Referee raised three major issues. Here are the individual responses to them:

2.1. Lack of information about the chronology

Response: This issue was also raised by Referee #1. We agree with both reviewers that more supporting information about the chronology of the record is desirable.

[Figure]

We have now added all the details of the radiocarbon dates, the estimated reservoir correction, and the age-depth curve in the form of a new table and figure in the Supplementary Material (see attached document). On the other hand, we tend to disagree with Referee #2 that an additional figure with the previous investigations of Lago Chungará should be added because those investigation were, in their majority, focused on different research problems and covered time periods not relevant for our manuscript.

2.2 More extended discussion of past ENSO variability

Response: We agree with Referee #2 that more records of past ENSO variations contribute to a more comprehensive discussion. Following the Referee's suggestion, we have added two of the four articles suggested by her/him. Added mentions to the ENSO reconstruction from Barr et al. (2019) (lines 394-396), and the Botriococcus spp. curves from El Junco Lake (Zhang et al., 2014) (lines 392-394), the latter record complements well with the evidence from the same lake presented in our original manuscript (Conroy et al., 2008). Despite of considering the two other suggested articles as relevant ENSO reconstructions, we decided not to include the Ortega et al., (2019) article suggested by Referee #2 because it provides evidence for ENSO variability at timescales ranging from multi-millennial to millennial. As such, its potential contribution to our discussion is not critical. Similarly, we decided not to include Vargas et al. (2006) because this article presents a chronology for individual ENSO-driven events that does not represent a continuous record able to capture mean-state values and/or centennial-scale anomalies of past ENSO activity. Nonetheless, we would like to stress that we consider these two articles as scientific relevant reconstructions.

2.3 ITCZ influence over SASM

Response: Referee #2 contents that we are too easily disregarding shifts in the ITCZ as potential drivers of SASM changes. We acknowledge that Referee # 2 has raised a relevant topic that demands a careful discussion. In our original manuscript we used the flagship Holocene record of the ITCZ position from the Cariaco Basin (% of Titanium; Haug et al., 2001). This record does not exhibit any noticeably centennial-scale excursion for the period between 2300-1000 cal yr BP (Fig. 06). Hence, we posit that our Chungará record was largely unaffected by centennial-scale north-south shifts of the ITCZ. Referee #2 suggested to analyse the Cariaco curve in the same way than Salvatecci et al. (2014), which examined this record by normalizing it with the mean value of the last 1650 years (Fig. 3 in that article). We note that this interval does not cover the main climate anomalies detected at Chungará (2600-1000 cal yr BP), which prevents a comprehensive comparison. Nonetheless, the Salvatecci et al. (2014) normalized curve shows overall positive values between 1650-1230 cal yr BP (300-720 AC in Fig. 3 from that article). These positive values could be understood as an overall northward position of the ITCZ relative to the mean state of the 1650 years, a change that could explain the dry anomaly observed at Chungará during that time. Nonetheless, we note the positive values of the Salvatecci et al. (2014) curve are only anomalous in the context of the last 1650 cal yr BP, being indistinguishable in a longer-term context. These anomalies result from the %Ti experiencing the lowest values of the entire Holocene during the Little Ice Age between 430-180 cal yr BP (Fig 3 in Haug et al., 2001). In other words, the anomalous positive values result largely from the mean normalization rather than a genuine northward shift in the ITCZ position during that time. To clarify our point, we have plotted the % of Titanium from Cariaco normalized by the mean of the last 1650 years (upper plot in attached figure) along with with the same proxy normalized by the mean 4500 years (lower plot). When normalizing by the mean values of the last 4500 years, any positive anomaly between 1650-1230 cal yr BP disappears. For all this reasons, we decide not to comment the Salvatecci et al. (2014) article and keep with our interpretation based on the raw Cariaco data presented in Haugh et al 2001. However, we are open to consider commenting on the Salvatecci et al (2014) article in the manuscript if the Editor consider relevant to do so.

Minor comment: Line 76-77. It is necessary to add updated references on the geological setting. Other authors (Barnes and Ehlers, 2009; Jordan et al., 2010) point out that slow steady-state uplift since 40 Ma is more consistent than enhanced short periods

of uplift. Response: We have added that a slow and more steady uplift has also been proposed, citing Barnes and Ehlers (2009).

Line 216. ": : :is supported by multiple lines of evidence". This sentence is too imprecise. References are needed. Response: The different types of evidence supporting the reservoir effect of the article are discussed at Giralt et al. (2008), as mentioned in the main text.

Line 355. What kind of geochemical data?. As it is mentioned above, a figure with the available data (such as Bao et al., 2015) is necessary. Response: By "geochemical data" we refer to the work of Giralt et al. (2008), which uses mineralogical and chemical parameters such as Total Organic Carbon, Total Biogenic Silica, XRF scanning and gray values among others. These analyses are now mentioned in section 2.3. and a reference to the study has been added following the lines mentioned by Referee #2.

Bao, R., Hernández, A., Sáez, A., Giralt, S., Prego, R., Pueyo, J., Moreno, A., and Valero-Garcés, B. L.: Climatic and lacustrine morphometric controls of diatom paleoproductivity in a tropical Andean lake, Quaternary Science Reviews, 129, 96-110, 2015. Barnes, J., and Ehlers, T.: End member models for Andean Plateau uplift, Earth-Science Reviews, 97, 105-132, 2009.

Barr, C., Tibby, J., Leng, M., Tyler, J., Henderson, A., Overpeck, J., Simpson, G., Cole, J., Phipps, S., and Marshall, J.: Holocene el Niño–southern Oscillation variability reflected in subtropical Australian precipitation, Scientific reports, 9, 1627, 2019.

Conroy, J. L., Overpeck, J. T., Cole, J. E., Shanahan, T. M., and Steinitz-Kannan, M.: Holocene changes in eastern tropical Pacific climate inferred from a Galápagos lake sediment record, Quaternary Science Reviews, 27, 1166-1180, http://dx.doi.org/10.1016/j.quascirev.2008.02.015, 2008.

Giralt, S., Moreno, A., Bao, R., Sáez, A., Prego, R., Valero-Garcés, B. L., Pueyo, J. J., González-Sampériz, P., and Taberner, C.: A statistical approach to disentangle environmental forcings in a lacustrine record: the Lago Chungará case (Chilean Altiplano), Journal of Paleolimnology, 40, 195-215, 2008.

Haug, G. H., Hughen, K. A., Sigman, D. M., Peterson, L. C., and Röhl, U.: Southward migration of the intertropical convergence zone through the Holocene, Science, 293, 1304-1308, 2001.

Hernández, A., Bao, R., Giralt, S., Sáez, A., Leng, M. J., Barker, P. A., Kendrick, C. P., and Sloane, H. J.: Climate, catchment runoff and limnological drivers of carbon and oxygen isotope composition of diatom frustules from the central Andean Altiplano during the Lateglacial and Early Holocene, Quaternary Science Reviews, 66, 64-73, 2013.

Jomelli, V., Khodri, M., Favier, V., Brunstein, D., Ledru, M.-P., Wagnon, P., Blard, P.-H., Sicart, J.-E., Braucher, R., and Grancher, D.: Irregular tropical glacier retreat over the Holocene epoch driven by progressive warming, Nature, 474, 196, 2011.

Jordan, T. E., Nester, P. L., Blanco, N., Hoke, G. D., Dávila, F., and Tomlinson, A.: Uplift of the Altiplano‐Puna plateau: A view from the west, Tectonics, 29, 2010.

Marcott, S. A., Shakun, J. D., Clark, P. U., and Mix, A. C.: A reconstruction of regional and global temperature for the past 11,300 years, science, 339, 1198-1201, 2013.

Ortega, C., Vargas, G., Rojas, M., Rutllant, J. A., Muñoz, P., Lange, C. B., Pantoja, S., Dezileau, L., and Ortlieb, L.: Extreme ENSO-driven torrential rainfalls at the southern edge of the Atacama Desert during the Late Holocene and their projection into the 21th century, Global and Planetary Change, 175, 226-237, 2019.

Rabatel, A., Francou, B., Jomelli, V., Naveau, P., and Grancher, D.: A chronology of the Little Ice Age in the tropical Andes of Bolivia (16 S) and its implications for climate reconstruction, Quaternary Research, 70, 198-212, 2008.

Rabatel, A., Francou, B., Soruco, A., Gomez, J., Cáceres, B., Ceballos, J. L., Basantes, R., Vuille, M., Sicart, J. E., Huggel, C., Scheel, M., Lejeune, Y., Arnaud, Y., Collet, M.,

Condom, T., Consoli, G., Favier, V., Jomelli, V., Galarraga, R., Ginot, P., Maisincho, L., Mendoza, J., Ménégoz, M., Ramirez, E., Ribstein, P., Suarez, W., Villacis, M., and Wagnon, P.: Current state of glaciers in the tropical Andes: a multi-century perspective on glacier evolution and climate change, The Cryosphere, 7, 81-102, 10.5194/tc-7-81-2013, 2013.

Sáez, A., Giralt Romeu, S., Hernández Hernández, A., Bao Casal, R., Pueyo Mur, J. J., Moreno Caballud, A., and Valero Garcés, B. L.: Comment on" Climate in the Western Cordillera of the Central Andes over the last 4300 years", by Engel et al.(2014), Quaternary Science Reviews, 2015, vol. 109, p. 126-130, 2015.

Salvatteci, R., Gutiérrez, D., Field, D., Sifeddine, A., Ortlieb, L., Bouloubassi, I., Boussafir, M., Boucher, H., and Cetin, F.: The response of the Peruvian Upwelling Ecosystem to centennial-scale global change during the last two millennia, Climate of the Past, 10, 715-731, 2014.

Vargas, G., Rutllant, J., and Ortlieb, L.: ENSO tropical–extratropical climate teleconnections and mechanisms for Holocene debris flows along the hyperarid coast of western South America (17–24 S), Earth and Planetary Science Letters, 249, 467-483, 2006.

Vuille, M., Burns, S., Taylor, B., Cruz, F., Bird, B., Abbott, M., Kanner, L., Cheng, H., and Novello, V.: A review of the South American monsoon history as recorded in stable isotopic proxies over the past two millennia, Climate of the Past, 8, 1309-1321, 2012.

Zhang, Z., Leduc, G., and Sachs, J. P.: El Niño evolution during the Holocene revealed by a biomarker rain gauge in the Galápagos Islands, Earth and Planetary Science Letters, 404, 420-434, 2014.

Please also note the supplement to this comment:
https://www.clim-past-discuss.net/cp-2019-13/cp-2019-13-AC2-supplement.pdf

1650-1230 cal yr BP

Last 1650 yr normalized
Salvatecci et al. (2014)

North

↑

ITCZ

↓

South

North

↑

ITCZ

↓

South

Last 4500 yr normalized

Cal yr BP

**Fig. 1.** Figure to comment section 2.3 ITCZ influence over SASM

---

## Referee Report (RR1)

Review report cp-2019-13

I enjoyed reading the manuscript. It is a thoughtful and comprehensive study on a topic of great interest for Climate of the Past's readership. The comments made by the reviewers in the interactive open discussion have been carefully incorporated, especially how different climate forcings are correlated to climatic teleconnections and its drivers for explaining precipitation anomalies in the Lake Chungara pollen record. I'm not an expert on this topic and so I only have suggestions for expanding the strength of modern pollen-environmental relationships to better interpret the fossil record.

How well the fossil pollen assemblages are represented by the surface vegetation types? It would be good to compute an unconstrained multivariate ordination analysis (e.g PCA) projecting the core samples into the ordination defined by modern pollen assemblages. In fact, it'd be even more relevant to carry out an ordination analysis with CCA or RDA constrained to climatic data (i.e. from WorldClim). By doing so, precipitation and/or temperature controls on fossil assemblages shifts could be assessed in a more nuanced way.

Is possible to comment on the idea of Equatorial Atlantic sea-surface temperature gradient variability to explain extra tropical source of moisture in Lake Chungara precipitation anomalies? Also, to what extent is possible to identify change in seasonality of precipitation anomalies in the Lake Chungara record?

Minor changes:
Line 156: "a few years later"
Line 179: It seems a bit contradictory to claim that the climate history of Lake Chungara is not well understood after such a great summary of paleolimnological and paleoclimatological studies carried out in the region over the last 20 years.
Line 198: please indicate which statistical software was used to carry out the CONISS analysis
Line 227: please change ordination by classification
Line 323: please change "periphylic" to "periphytic" (diatoms)
Line 356: replace "in the core" by "in the central"

---

## Author Response (AR2)

Response to review report cp-2019-13 made by Referee #4

1. Use of multivariate statistics

Referee #4 suggested the use of multivariate statistics to explore the pollen-vegetation-climate relationships presented in our original manuscript. This represents a constructive suggestion that may ultimately help to strengths our paleoclimate inferences. Hence, we have followed referee's suggestions performing all the statistical analyses recommended by him/her. A description of the methods and results of such analyses can be found the supplementary information, while the discussion and implications of the results can be found at different parts of the corrected manuscript.

A brief response to all referees' specific comments regarding multivariate statistics are found below.

1.1 Referee #4 suggested to explore the underlying climate controls of our pollen data. Following this recommendation, we have performed a Canonical Correspondence Analysis (CCA) on our surface pollen dataset constrained to the climate variables of WorldClim (Fig. S2). Fig. S2 shows that all surface pollen samples are clustered following the regional altitudinal vegetation belt described in section 2.1. Hence, this analysis supports the close climate-pollen relationship reported previously in our study area. This same relationship also emerges from an inspection of Fig 3 of our original manuscript. Equally important, Fig. S2 shows a strong and equal-sign affinity between precipitation (both annual and DJFM), high-Andean taxa and pollen assemblages of surface samples above 4000 masl. These affinities support the close link between precipitation anomalies and the changes in high-Andean taxa observed in the Chungará record, and therefore references to CCA results were added to the corrected version of the manuscript in lines 270-273, 308-309 and 326-328.

1.2 Referee #4 also mentions "how well the fossil pollen assemblages are represented by the surface vegetation types?" To answers this question, we have projected the core samples into a Principal Component Analysis (PCA) defined by modern pollen assemblages (Fig. S3), as suggested by the referee. Fig. S3 reveals that the diversity of the surface pollen spectra is far larger than the fossil ones. This is expected because Lago Chungará pollen record reflects predominately vegetation changes occurring in the upper portion of the pollen transect and not across its entire elevational range. Similarly, fossil samples are clustered with the surface samples of the highest elevation, as Lago Chungará sits above the limit of the modern pollen transect. Similarly, the differences between the fossil assemblages corresponding to the humid and dry anomalies that are discussed in the manuscript are indistinguishable in the context of the entire diversity of the fossil + surface pollen spectra.

However, when projecting only the fossil assembles (Fig. S4), humid samples show closer affinities to the high-Andean steppe taxa and high lake level indicators, whereas dry samples are more closely associated with Prepuna and low lake stands indicators. We have referred to these results to support our interpretation in line 322-323 and 335-336 of the revise manuscript.

Overall, the results of Figures S2, S3 and S4 confirm our pollen-climate inferences and paleoclimatic interpretations regarding the humid and dry anomalies recorded in Lago Chungará.

2. Atlantic temperature gradient

Referee #4 suggested to comment of the idea of Equatorial Atlantic sea-surface temperature gradient to explain the precipitation anomalies observed in the Chungará record.

We consider this a constructive suggestion, and therefore we have commented on this specific issue in line 429-433 of the revised manuscript.

3. To what extent if tis possible to identify change in seasonality of precipitation anomalies in the Chungará record?

We have shown that summer precipitation (DJFM) in the Lago Chungará represents about 90% of the total precipitation, and therefore any hydrological anomaly will be almost certainly linked to summer rainfall. Thus, the Chungará record provided very little information about possible changes in the seasonal precipitation cycle. If they occurred, the impact over the Chungará record was -almost certainly- limited. We are unaware of any paleoclimate evidence indicating significant changes in the seasonal precipitation cycle in tropical South America over the most recent millennia. Significant changes in precipitation seasonality might be expected to have occurred at glacial-interglacial timescales, yet they are unlike to have occurred over the last 4000 years when modern global atmospheric circulation was well-stablished. Nonetheless, we are open to add additional comment on this issue if the referee of yourself provides with relevant information.

3. Minor changes

Line 156: "a few years later"

Response: Corrected

Line 179: It seems a bit contradictory to claim that the climate history of Lake Chungará is not well understood after such a great summary of paleolimnological and paleoclimatological studies carried out in the region over the last 20 years.

Response: We are cautioned to mention that the climate history of "the most recent millennia" is the one that has not been address in enough detail.

Line 198: please indicate which statistical software was used to carry out the CONISS analysis

Response: added required information

Line 227: please change ordination by classification

Response: change ordination by "cluster analysis"

Line 323: please change "periphylic" to "periphytic" (diatoms)

Response: Changed

Line 356: replace "in the core" by "in the central"

Response: Changed

[revised manuscript text omitted]